# CRIMP: a CRISPR/Cas9 insertional mutagenesis protocol and toolkit

Lee B. Miles [1], Vanessa Calcinotto [1], Sara Oveissi[1], Rita J. Serrano [1], Carmen Sonntag [1], Orlen Mulia [1], Clara Lee[1] & Robert J. Bryson-Richardson [1] ✉

Site-directed insertion is a powerful approach for generating mutant alleles, but low efficiency and the need for customisation for each target has limited its application. To overcome this, we developed a highly efficient targeted insertional mutagenesis system, CRIMP, and an associated plasmid toolkit, CRIMPkit, that disrupts native gene expression by inducing complete transcriptional termination, generating null mutant alleles without inducing genetic compensation. The protocol results in a high frequency of integration events and can generate very early targeted insertions, during the first cell division, producing embryos with expression in one or both halves of the body plan. Fluorescent readout of integration events facilitates selection of successfully mutagenized fish and, subsequently, visual identification of heterozygous and mutant animals. Together, these advances greatly improve the efficacy of generating and studying mutant lines. The CRIMPkit contains 24 ready-to-use plasmid vectors to allow easy and complete mutagenesis of any gene in any reading frame without requiring custom sequences, modification, or subcloning.

The advent of CRISPR/Cas9 technology has revolutionised functional genetics by enabling the generation of targeted double-stranded breaks. This breakthrough has facilitated precise site-directed insertional mutagenesis of genes, offering unprecedented precision. However, despite its transformative potential, this technique has historically faced challenges that hinder its widespread adoption. These challenges include low targeting efficiencies, the necessity of specialised technical expertise, and the requirement for custom targeting vectors tailored to each specific target site. As a result, there is a pressing need for optimised resources that enhance efficiency and universal genetic tools that can be seamlessly applied to any gene of interest.

Standard mutagenesis approaches use CRISPR/Cas9 to generate indels that result in a loss of function allele, however, the introduction of nonsense mutations from the insertion of stop cassettes or generation of indel mutations has the potential to induce genetic compensation[1–3], masking loss of function phenotypes. Additionally, the efficiency of editing can vary dramatically depending on the complexity of the modification, and therefore substantial time and cost can be involved in genotyping and identifying founders. Targeted integration has the potential to overcome both of these issues, and site-directed integration of foreign sequence through homology-directed repair (HDR) has been successful, with inserts ranging from simple DNA oligonucleotides[4] to entire expression cassettes[5–7]. However, several drawbacks, such as requiring a custom synthesis of a targeting vector for each target gene, and low integration efficiencies, have limited the use of HDR. Targeted integration via non-homologous end-joining (NHEJ), is more efficient than HDR but more error-prone, making it less appealing for targeting coding sequences. However, an intron-targeting strategy where a splice acceptor and downstream coding sequence are inserted into the intronic region of a gene, truncating the protein, overcomes this issue and has been used to generate mutant alleles[8–11]. Although more efficient than HDR, the NHEJ-intron-targeting approach has previously required individual targeting vectors to be cloned for each gene, limiting its broad application.

[1]School of Biological Sciences, Monash University, Clayton, Melbourne, VIC 3800, Australia. ✉e-mail: robert.bryson-richardson@monash.edu

To overcome these issues and enable a broader uptake of NHEJ-mediated insertional mutagenesis we describe a universal and highly efficient targeted knock-in mutagenesis system and toolkit that disrupts native gene expression by inducing complete transcriptional termination to produce full mutagenesis without genetic compensation and demonstrate its use in zebrafish.

Here we describe the CRISPR/Cas9 Insertional Mutagenesis Protocol (CRIMP) and toolkit (CRIMPkit) containing 24 ready-to-use plasmid vectors to allow easy and complete mutagenesis in any reading frame. Importantly, these vectors are universal, requiring no customisation of the sequence, modification, or subcloning. All CRIMPkit vectors are available from Addgene as a complete kit (#1000000225) or individual plasmids (199469-199492). Once integrated, these vectors drive a fluorescent reporter under the control of the targeted gene promoter, disrupting native gene expression, provide a reporter of the targeted genes expression pattern, and allow visual identification of successfully mutagenized fish. The toolkit contains multiple versions of each plasmid with different fluorophores; *mTagBFP2, mKate2*, and the *splitGFP* system to visually genotype fish and facilitate automated high-throughput screening, as well as multiple reporter expression systems enabling visual selection of fluorophores when expressed from either high or low-expressed target genes.

## Results

### CRISPR/Cas9 insertional mutagenesis protocol design
To achieve our goal of efficient mutagenesis, we developed the CRISPR/Cas9 Insertional Mutagenesis Protocol (CRIMP). We modified an existing zebrafish CRISPR/Cas9 injection protocol available from Integrated DNA technologies[TM] (IDT). Our protocol differs by using a higher concentration of guideRNAs[12] and Cas9 protein (Cas9 HiFi V3 (IDT)), the addition of KCL to improve solubility[13], and includes a targeting plasmid vector. GuideRNA molecules are well tolerated even when injected at high concentrations, however, the contaminates resulting from the purification of in-house synthesised guideRNAs, often cause toxicity when injected at high concentrations[12]. We, therefore, used the two-part Alt-R[TM] crRNA and tracrRNA system (IDT[TM]), which are complexed to generate guideRNAs. These are commercially prepared and chemically modified and, in our experience, have high-efficiency cutting, are resistant to RNase degradation, and do not cause toxicity when injected at high concentrations (>300 ng/µl).

To achieve integration as rapidly as possible after injection, we provide Cas9 as a protein rather than an mRNA. Zebrafish have a pause phase in the cell cycle during the first cell division[14], which results in an extended cell cycle length during this first cleavage event[15], after which they undergo extremely rapid cell divisions, which continues until the midblastula-transition where G2 cell cycle pauses are acquired[16] and the zygotic genome becomes active. We reasoned that the lag in Cas9 production when injecting *Cas9* mRNA would miss this early pause phase and result in lower integration efficiency. In order to facilitate early integration events during this first cell division, we used pre-complexed guideRNA/Cas9 ribonucleoprotein (RNP) and targeting plasmid in our injection mix. We aimed to collect embryos within 5 min of fertilisation and perform injections within 15 min to ensure the highest likelihood for early integration events during this first cell division.

### CRIMP optimisation
To test the ability of CRIMP to efficiently generate targeted insertions, we selected *actc1b* as a target gene as it is expressed in the skeletal muscle at high levels (5384 transcripts per million (TPM) at 24 hpf[17]), facilitating visualisation of integration events. We inserted a targeting vector with a splice acceptor site followed by *mTagBFP2* into intron 2 of *actc1b*, to generate the targeted insertion line *Ti(actc1b^int2^-mTagBFP2)* (Fig. 1a, b, c, f), hereafter referred to as *actc1b^mTagBFP2^*. Of the

injected embryos, 15% (22/148) displayed mTagBFP2 expression and, surprisingly, we observed two embryos (1.3%) with fluorescence throughout one half of their body (Fig. 1e and Table 1), indicating that integration occurred rapidly after injection, either during or directly after the first cell division. To our knowledge, such rapid early integration events during the first cell division have not been previously reported for targeted insertions and represent a substantial improvement in the timing of integration using our protocol compared to previously reported methods. These embryos were raised, and the outcrossing of an individual half body plan-positive fish produced offspring with the correct fluorescent expression pattern in 38% of progeny (21/54, Supplementary Table 1).

We are interested in high-throughput drug screening and the ability to visually select homozygous mutants to increase throughput and reduce costs. To enable this and further test the CRIMP approach, we incorporated the splitGFP system[18] into the inserted vector. We inserted a vector containing a splice acceptor followed by *mTagBFP2*, a T2A cleavage sequence[19], and then one half of splitGFP (*pSA0-mTagBFP2-T2A-splitGFP1-10*) into intron 2 of *actc1b*, to generate the *Ti(actc1b^int2^-mTagBFP2-T2A-splitGFP1-10)* transgenic insertion line (Fig. 1c, d, g), hereafter referred to as *actc1b^sGFP1^*. 15.1% of injected embryos had mTagBFP2 expression, and two (2.2%) injected embryos displayed the correct expression pattern in one half of the embryo body plan (Table 1). A second allele was generated by inserting the *pSA0-mTagBFP2-T2A-splitGFP11x7* vector into intron 4 of *actc1b*, to generate the *Ti(actc1b^int4^-mTagBFP2-T2A-splitGFP11x7)* transgenic line (Fig. 1c, d, g), hereafter referred to as *actc1b^sGFP2^*. Of the injected embryos, 11.4% (43/377) displayed mTagBFP2 expression, and one embryo (0.2%) displayed the correct expression in half of the embryo body plan (Table 1). We suspect that the lower efficiency of half body plan-positive embryos in this experiment was due to the time taken to inject a large number of embryos (*n* = 377, Table 1), which may have resulted in a large proportion being injected during the later stages of the first cell division, missing the optimal time window for early integration during or before the first cell division. Therefore, we suggest injecting small batches of very recently laid embryos at a time to promote integration events before the cell completes its first cell division. Embryos displaying expression in half the body plan were raised to adulthood and outcrossed to wildtype fish to identify founders. All individuals that displayed such expression demonstrated germline transmission of the targeted insertion with the correct fluorophore expression pattern corresponding to *actc1b*.

Crossing *actc1b^sGFP1^* and *actc1b^sGFP2^* lines together at the F2 generation resulted in non-fluorescent wildtype embryos, heterozygous embryos that are mTagBFP2 positive, but negative for GFP fluorescence, and *actc1b* mutant embryos (compound heterozygotes) that can be identified by GFP fluorescence (Fig. 1g) in addition to mTagBFP2.

### Integration can be detected in genes with low expression levels
Our initial lines targeted *actc1b* which has very high levels of expression, not representative of most genes. To demonstrate the use of CRIMP to generate mutant alleles in genes with moderate expression levels we chose *bag3* as a target gene which is broadly expressed in skeletal muscle at 30 TPM at 24 hpf[17] and has a well-established mutant phenotype[20]. We targeted a vector with a splice acceptor site and *mTagBFP2* to intron 2 of *bag3* to generate the targeted insertion line *Ti(bag3^int2^-mTagBFP2)* (Fig. 2), hereafter referred to as *bag3^mTagBFP2^*.

In this experiment we used freshly prepared guideRNA complexes and increased the duration of the incubation of guideRNAs, Cas9 protein, and targeting plasmid to 30 min (Table 1), in order to allow greater Cas9 binding of the targeting plasmid[21], which we reasoned would increase the rate of mosaic integration. Positive mTagBFP2 fluorescence was observed in 75% (90/120) of the injected embryos (Table 1), representing a dramatic improvement in the frequency of

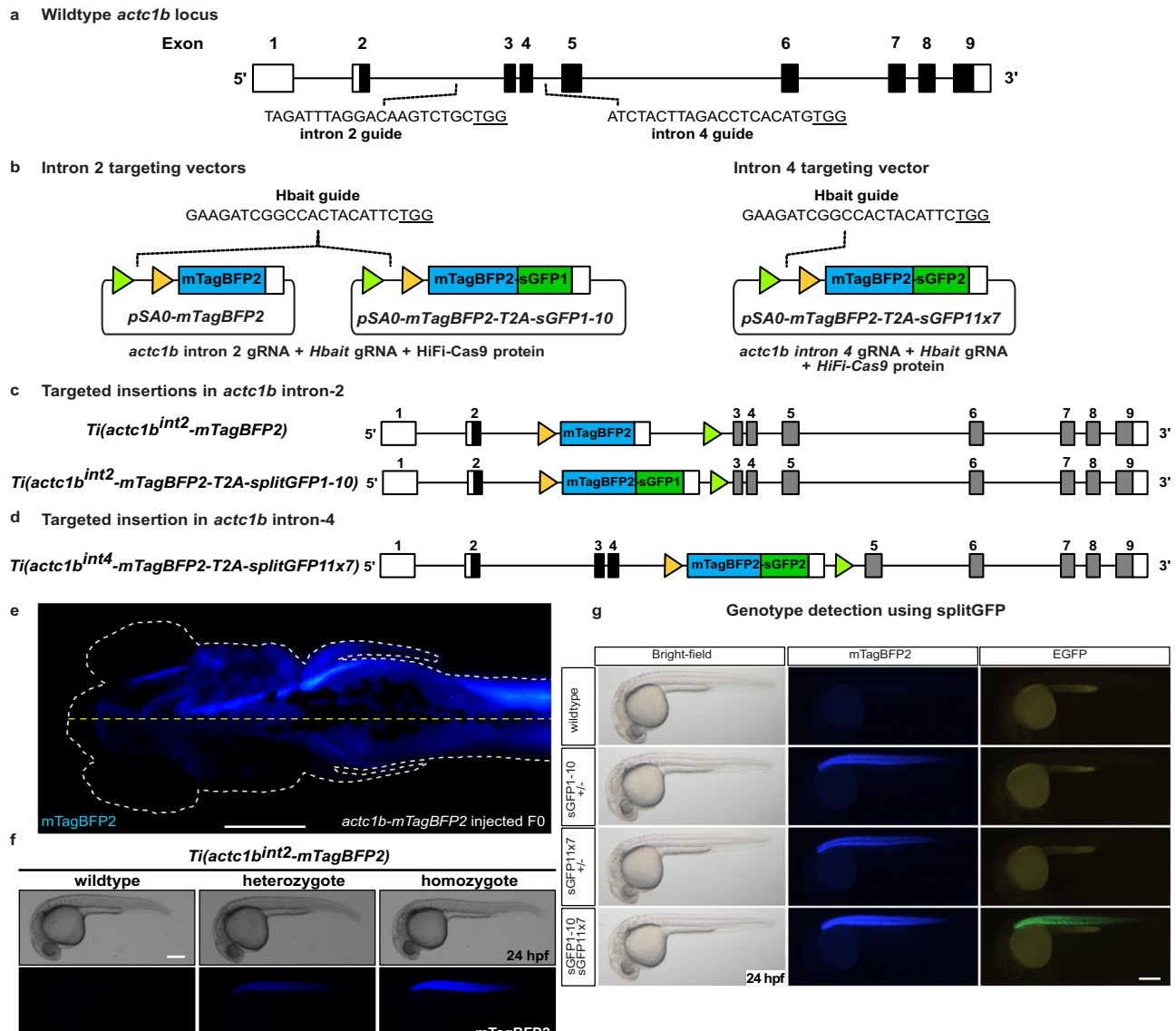

**Fig. 1 | SplitGFP alleles enable visual selection of homozygous mutants.**
**a** Schematic of *actc1b* gene structure, and location of the intron 2 and intron 4 guideRNA sites. **b** Schematic of intron 2 and intron 4 targeting vectors, FRT and FRT3 sites are indicated by triangles. **c** Schematic of *Ti(actc1b^int2^-mTagBFP2)* and *Ti(actc1b^int2^-mTagBFP2-T2A-sGFP1-10)* intron 2 targeted insertion lines. **d** Intron 4 targeted *Ti(actc1b^int4^-mTagBFP2-T2A-sGFP11x7)* insertion line. **e** Example of successful targeted integration during the first cell division. Integration of the targeting vector into intron 2 of *actc1b* during the first cell division results in embryos with mTagBFP2 expression in one half of the ventral body plan. A white dotted line

indicates embryo outline. The yellow dotted line indicates the ventral axis. In our experience, such embryos, when raised to adulthood and crossed, have always transmitted the successful integration event to their progeny. Ventral view of a 4 dpf embryo. **f** Brightfield and fluorescent images of wildtype, heterozygote, and homozygote embryos at 24 hpf. **g** 24 hpf embryos were generated by crossing the two *actc1b^sGFP^* lines together. Heterozygous carriers display mTagBFP2 expression only while, in addition to mTagBFP2 expression, compound heterozygous mutants can be visually selected by GFP fluorescence. Experiments were repeated three independent times. Scale bar = 250 μm.

integration events, and therefore this longer incubation was retained for all subsequent experiments. We additionally tested this improvement using the two *actc1b* targeting guideRNAs described above and saw dramatic improvements in integration (improving from 11 and 15% integration events to 57 and 67%, respectively, and examples of fish with either half or the full body plan expression; Supplementary Table 2 and Fig. 1), alleviating any concerns that the improvement may be due to the guideRNA rather than the protocol. Four embryos with high proportions of mTagBFP2 expressing cells were raised to adulthood and outcrossed to wildtype fish. One of these successfully transmitted the targeted insertion to its offspring to generate a stable line (25%) (Table 1). Heterozygous F2 fish were crossed together, and the embryos were subjected to a 1% methylcellulose assay at 24 hpf[20]. Wildtype and heterozygous embryos were phenotypically normal after

a methylcellulose assay treatment; however, homozygous mutants display broken fibres phenocopying the published *bag3* mutant phenotype (Fig. 2d). qRT-PCR analysis identified that the *bag3* transcript is lost in *bag3^mTagBFP2^* homozygotes (Fig. 2e). *mTagBFP2* was detected at similar levels to native *bag3* expression (Fig. 2f).

We next selected *tdgf1* and *vegfaa* as target genes, as they have low levels of expression (*tdgf1* 10 TPM and *vegfaa* 9 TPM at 24 hpf[17]), restricted expression patterns, very well-characterised mutant phenotypes[22–25], and *vegfaa* mutants have previously been demonstrated to display genetic compensation[1,3].

Targeting of a splice acceptor-*mTagBFP2* vector into intron 3 of *tdgf1* did not result in any detectable mTagBFP2 fluorescence in the injected embryos, which we suspected was due to the low expression levels of the *tdgf1* gene. To overcome this, we generated Gal4/UAS

**Table 1 | Integration efficiency and founder screening**

| Injection efficiency | | | | | Founder screening | | | |
|---|---|---|---|---|---|---|---|---|
| Target gene-construct | Injected | Mosaic | Half body plan | Total positive | | Raised | Screened | Positive |
| actc1b-mTagBFP2-afpUTR | 148 | 20 (13.4%) | 2 (1.3%) | 22 (15%) | Mosaic | 20 | 20 | 0 (0%) |
| | | | | | Half body plan | 2 | 1 | 1 (100%) |
| | | | | | Total | 22 | | |
| actc1b-mTagBFP2-sGFP1-10 | 93 | 12 (12.9%) | 2 (2.2%) | 14 (15.1%) | Mosaic | 2 | 2 | 0 (0%) |
| | | | | | Half body plan | 1 | 1 | 1 (100%) |
| | | | | | Total | 3 | | |
| actc1b-mTagBFP2-sGFP11x7 | 377 | 42 (11%) | 1 (0.2%) | 43 (11.4%) | Mosaic | 5 | 4 | 1 (25%) |
| | | | | | Half body plan | 1 | 1 | 1 (100%) |
| | | | | | Total | 11 | | |
| **(Protocol optimisation: Freshly prepared guideRNA & >30 min incubation at 37 °C)** | | | | | | | | |
| bag3-mTagBFP2-afpUTR | 120 | 90 (75%) | - | 90 (75%) | Mosaic | 4 | 4 | 1 (25%) |
| | | | | | Half body plan | - | - | - |
| | | | | | Total | 4 | | |
| tdgf1-Gal4-afpUTR | 136 | 128 (94.1%) | 1 (0.7%) | 129 (94.9%) | Mosaic | 8 | 0 | 0 |
| | | | | | Half body plan | 1 | 1 | 1 (100%) |
| | | | | | Total | 9 | | |
| vegfaa-Gal4-afpUTR | 180 | 172 (95.6%) | 1 (0.6%) | 173 (96.1%) | Mosaic | 11 | 1 | 0 |
| | | | | | Half body plan | 1 | 1 | 1 (100%) |
| | | | | | Total | 12 | | |
| vegfaa-Gal4-synCoTC | 398 | 378 (95%) | - | 378 (95%) | Mosaic | 11 | 6 | 3 (50%) |
| | | | | | Half body plan | - | - | - |
| | | | | | Total | 11 | | |

The integration efficiency of each target site in $F_O$ embryos is located on the left side of the table. The details of the founder screening are located on the right side of the table.

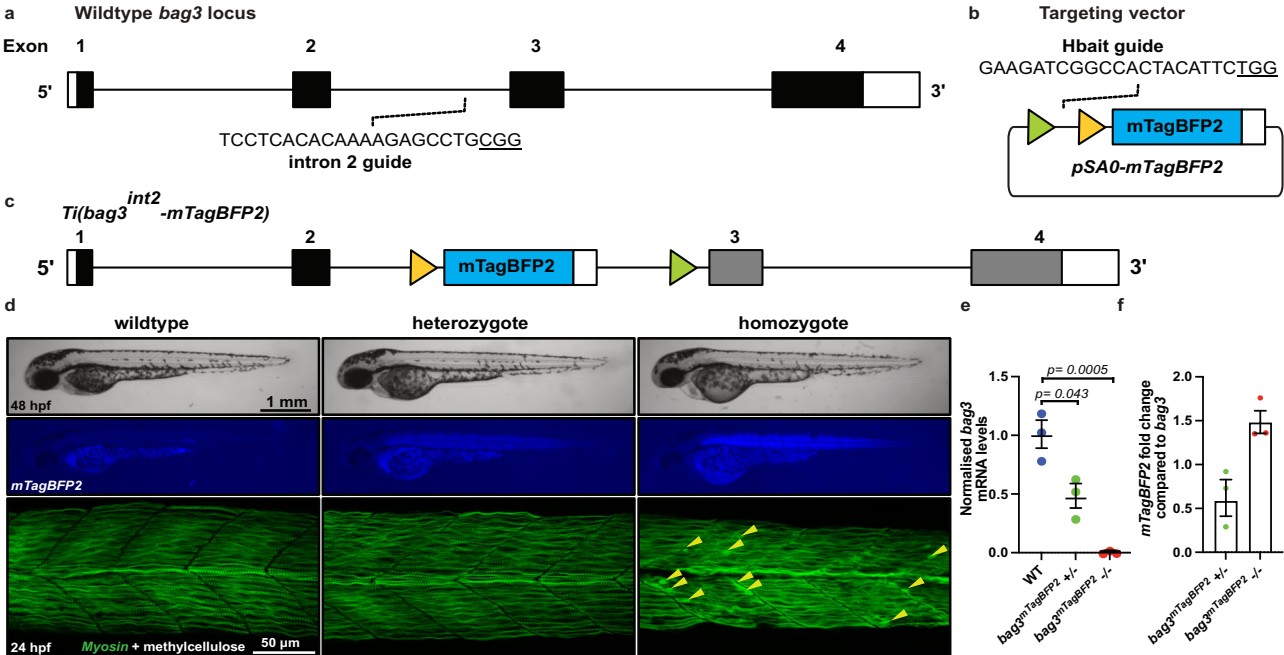

**Fig. 2 | Site-specific integration of targeting vector into *bag3*. a** Schematic of *bag3* gene structure, and location of the intron 2 guideRNA site. **b** Schematic of pSA0-mTagBFP2 targeting vector, FRT and FRT3 sites are indicated by triangles. **c** Schematic of *Ti(bag3^{int2}-mTagBFP2)* targeted insertion line. **d** Brightfield and fluorescent images of wildtype, heterozygote, and homozygote embryos at 24 hpf. Confocal images demonstrating loss of *bag3* results in broken muscle fibres (yellow arrows) in homozygote mutants at 24 hpf after a 1 h 1% methylcellulose treatment.

**e** mRNA expression analysis of *bag3* demonstrates a complete loss of native *bag3* transcript in homozygote mutants. **f** Fold change of *mTagBFP2* mRNA levels in *bag3^{mTagBFP2}* heterozygous and homozygous embryos compared to *bag3* levels in wildtype siblings. Error bars represent SEM for $n = 3$ independent biological replicates, each consisting of a pooled sample of 16 or 19 embryos. *rpl13* and *ef1α* were used as the reference genes. Source data are provided as a Source Data file. Statistical differences were determined using a two-tailed unpaired *t*-test.

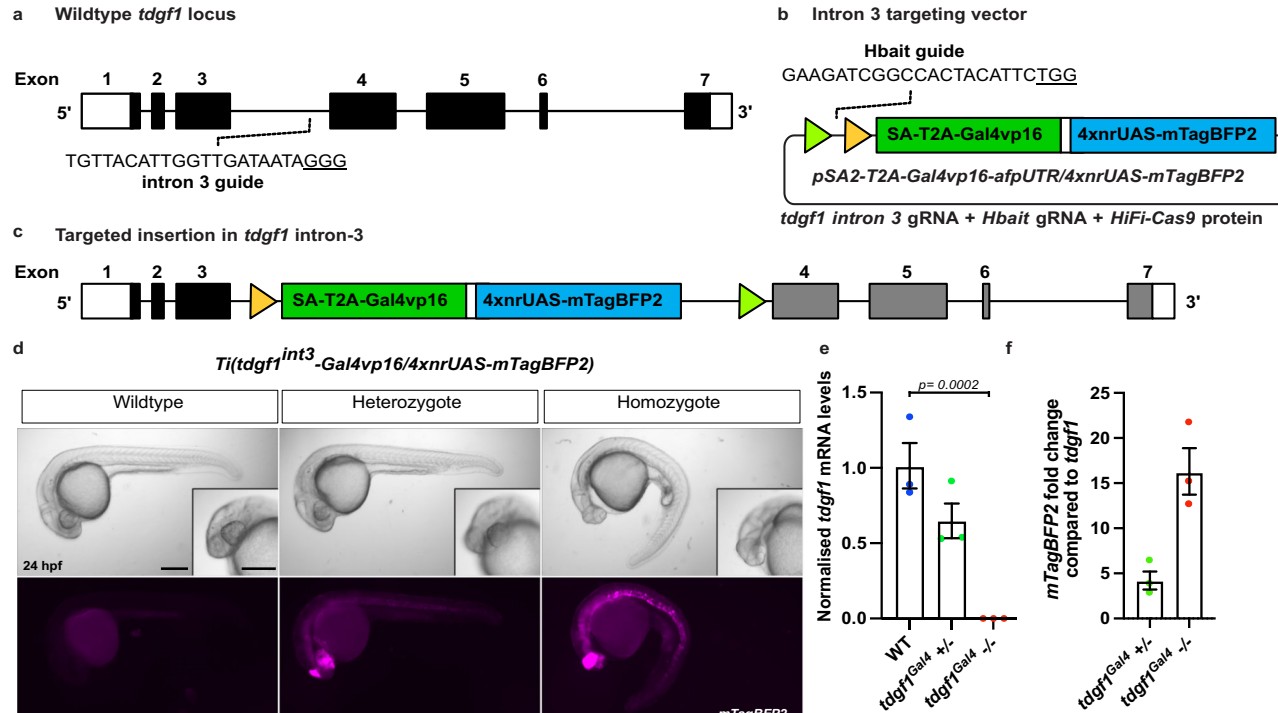

**Fig. 3 | Site-specific integration of targeting vector into *tdgf1*. a** Schematic of *tdgf1* gene structure and location of the intron 3 guideRNA target site. **b** The pSA2-*Gal4vp16/4xnrUAS-mTagBFP2* targeting vector, FRT and FRT3 sites are indicated by triangles. **c** Successful integration of the targeting plasmid to generate the *Ti(tdgf1^int3^-Gal4vp16/4xnrUAS-mTagBFP2)* targeted insertion line. **d** Brightfield and fluorescent images of *Ti(tdgf1^int3^-Gal4vp16/4xnrUAS-mTagBFP2)* targeted insertion line demonstrating loss of *tdgf1* phenotypes in homozygote mutants at 24 hpf.

mTagBFP2 fluorescence is false-coloured magenta. Scale bars = 250 μm. **e, f** mRNA expression analysis at 24 hpf demonstrates loss of *tdgf1* in homozygote mutants (**e**) and the fold increase of *mTagBFP2* expression levels compared to native *tdgf1* from Gal4/UAS amplification (**f**). Error bars represent SEM for $n = 3$ independent biological replicates, each consisting of a pooled sample of 14 or 16 embryos. *rpl13* and *ef1α* were used as the reference genes. Source data are provided as a Source Data file. Statistical differences were determined using a two-tailed unpaired *t*-test.

amplifying expression vectors. We inserted a splice acceptor-*T2A-Gal4vp16/4xnrUAS-mTagBFP2* vector into intron 3 of *tdgf1*, to generate the targeted insertion line *Ti(tdgf1^int3^-Gal4vp16/4xnrUAS-mTagBFP2)* (Fig. 3). Of the injected embryos 94.9% (129/136) had detectable mosaic mTagBFP2 expression and included one embryo (0.7%) displaying the correct expression pattern throughout half of the body plan (Table 1). When raised to adulthood and crossed to wildtype, this individual passed on the transgenic insertion, successfully establishing a *tdgf1* mutant line (Table 1). Heterozygous carriers are phenotypically wildtype with mTagBFP2 expressed in the same pattern as the published expression data for *tdgf1*[24]. Embryos homozygous for the insertion phenocopy *tdgf1* mutants displaying eye and head defects and a ventrally curved body (Fig. 3) as previously reported[22–24]. qRT-PCR analysis of transcripts in this line identified no detectable *tdgf1* transcript in homozygous mutants (Fig. 3e). In heterozygous and homozygous embryos, the Gal4/UAS expression system resulted in *mTagBFP2* transcripts at 4- and 16-fold higher levels than *tdgf1* in wildtype embryos respectively (Fig. 3f).

Similar to the results for *tdgf1*, targeting intron 1 of *vegfaa* with a splice acceptor-*mTagBFP2* vector did not result in any detectable mTagBFP2 fluorescence in the injected embryos. We inserted the splice acceptor-*T2A-Gal4vp16/4xnrUAS-mTagBFP2* vector into intron 1 of *vegfaa*, to generate the *Ti(vegfaa^int1^-Gal4vp16/4xnrUAS-mTagBFP2)* targeted insertion line (Fig. 4), hereafter referred to as *vegfaa^afpUTR^*. Similar to *tdgf1*, 96% of injected embryos had detectable mosaic mTagBFP2 expression, including one embryo (0.6%) displaying the expected expression pattern in half of the body plan which was raised, and successfully passed on the targeted transgenic insertion (Table 1). Heterozygous *vegfaa^afpUTR^* embryos were phenotypically wildtype and homozygous embryos displayed blood pooling, pericardial oedema, and no circulation (Fig. 4e). Crossing the line to a *Tg(kdrl:EGFP)*

reporter line allowed us to further identify that *vegfaa* mutant animals lacked a dorsal aorta, and displayed a disruption of intersegmental vessel branching in the tail (Fig. 4g), consistent with published *vegfaa* mutants[3,25]. qRT-PCR analysis in this line identified that *mTagBFP2* transcripts were found at levels 2 fold higher in *vegfaa^afpUTR^* heterozygotes and 7.3-fold higher in homozygotes compared to *vegfaa* expression levels in wildtype siblings (Fig. 4f), v*egfaa* transcripts were lost in homozygotes (Fig. 4h).

## Transcriptional termination reduces variability in reporter expression

We noticed that mTagBFP2 expression levels were often variable in *vegfaa^afpUTR^* fish (Supplementary Fig. 2). We reasoned that the reporter variation might be due to incomplete transcriptional termination of the *Gal4vp16* interfering with the downstream UAS expression cassette.

To test this we incorporated a Co-transcriptional Cleavage (CoTC)-type terminator sequence to prevent read-through of the RNA polymerase II[26]. We generated a synthetic Co-transcriptional Cleavage (CoTC)-type terminator element, termed synCoTC, consisting of the afpUTR up until the poly-A signal, followed by the human *CCNB1* CoTC[26]. We then generated a second *vegfaa* insertion line with the *pSAO-T2A-Gal4vp16_synCoTC/4xnrUAS-mTagBFP2* vector (Fig. 4c, f), hereafter referred to as *vegfaa^synCoTC^*). 95% of injected embryos had detectable mosaic mTagBFP2 expression, and six embryos displaying the highest level of mosaicism were raised to adulthood and screened, three (50%) of which passed on the insertion to progeny to generate founder lines (Table 1). We did not detect any embryos with expression throughout one half of the body plan in this injection, however, this might have been due to a delay in injection timing, potentially missing the time window required for early integration during the G2 phase of

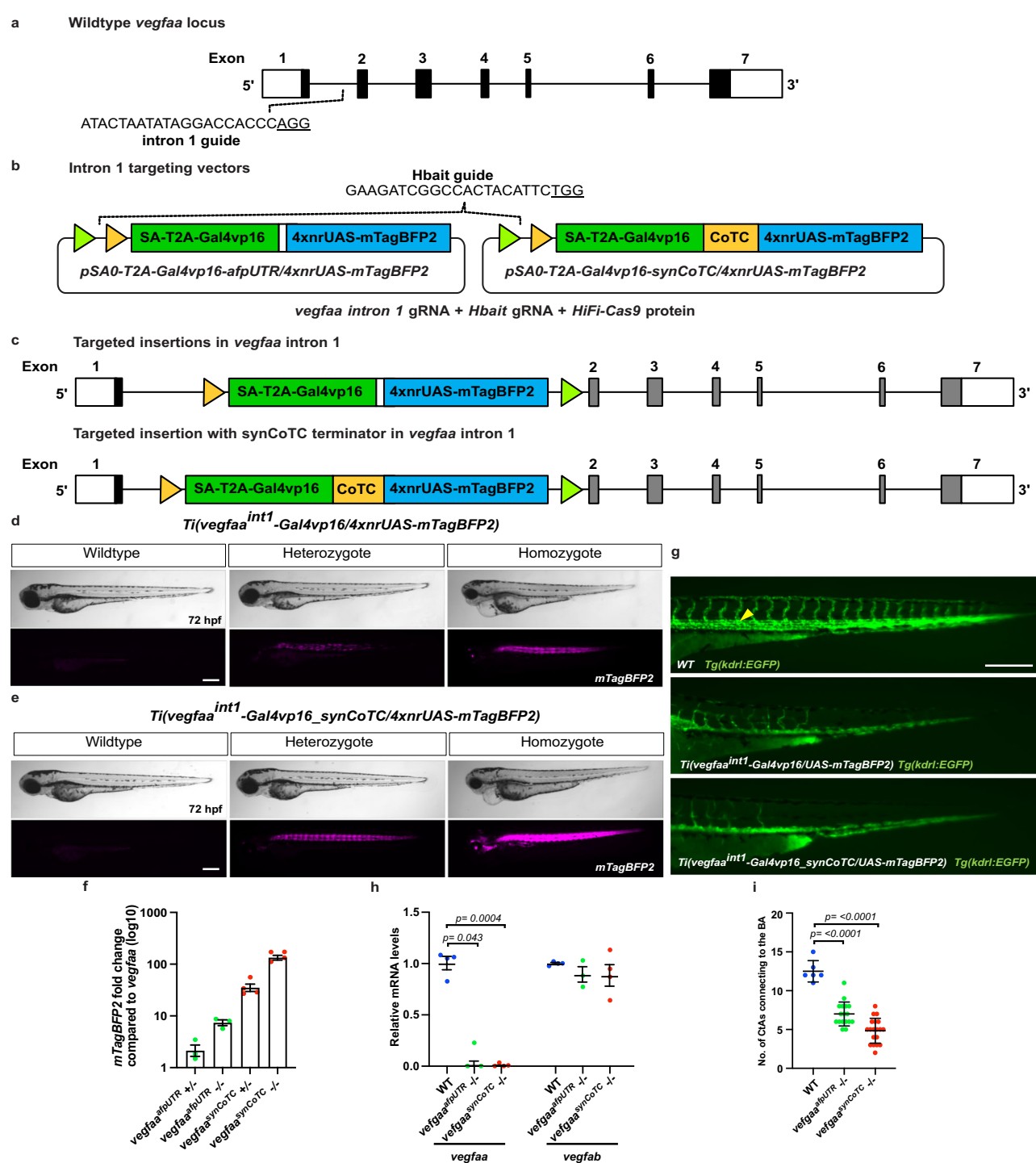

the first cell division. There was no difference in the homozygous mutant phenotype of the *vegfaa*^synCoTC line compared to the no-terminator line *vegfaa*^afpUTR (Fig. 4e, g); however, the mTagBFP2 expression was much brighter and more robust, without variation in the mTagBFP2 levels (Fig. 4e). Gene expression analysis identified loss of *vegfaa* transcripts in *vegfaa*^synCoTC mutants (Fig. 4h). The Gal4/UAS expression system amplified *mTagBFP2* levels 35 fold higher in heterozygotes, and 133 fold higher in homozygotes, compared to wildtype *vegfaa* expression levels (Fig. 4f). This is considerably higher than in the *vegfaa*^afpUTR line and demonstrates that inclusion of the synCoTC terminator can act to prevent variable expression and produces insertional lines with higher levels of reporter fluorescence.

Premature stop mutations prior to the last exon in *vegfaa* have been shown to induce genetic compensation, resulting in an upregulation of the paralogous gene *vegfab*[1,3]. The non-compensating promoter deletion mutant of *vegfaa* displays a stronger intracerebral central arteries (CtAs) branching defect than *vegfaa* compensating mutants[3]. Gene expression analysis in our CRIMP mutants identified a loss of *vegfaa* expression without upregulation of *vegfab* (Fig. 4h). Quantification of CtA branching in our CRIMP mutants identified a reduction in the branching of the CtAs (Fig. 4i) at a similar level to that of the non-compensating promoterless *vegfaa* mutants[3]. Together these results demonstrate that, as expected, CRIMP insertional mutants do not undergo genetic compensation.

**Fig. 4 | Site-specific integration of targeting vectors into *vegfaa*. a** Schematic of *vegfaa* gene structure, location of the intron 1 guideRNA target site. **b** The targeting vectors pSA2-*Gal4vp16/4xnrUAS-mTagBFP2* and pSA2-*Gal4vp16_synCoTC/4xnrUAS-mTagBFP2*, FRT and FRT3 sites are indicated by triangles. **c** Successful integration of the targeting plasmid to generate the *Ti(vegfaa^{int1}-Gal4vp16/4xnrUAS-mTagBFP2)* hereafter referred to as *vegfaa^{afpUTR}* and *Ti(vegfaa^{int1}-Gal4vp16_synCoTC/4xnrUAS-mTagBFP2)* hereafter referred to as *vegfaa^{synCoTC}* targeted insertion lines. **d, e** *vegfaa* targeted insertion lines demonstrating loss of *vegfaa* phenotypes in homozygote mutants at 48 hpf. **d** *vegfaa^{afpUTR}* transgenic insertion line. Scale bar 250 µm. **e** *vegfaa^{synCoTC}* transgenic insertion line. mTagBFP2 fluorescence (magenta) in d and e was captured with the same imaging settings, demonstrating more robust expression levels in the synCoTC-containing line. Scale bar 250 µm. **f** mRNA levels at 24 hpf demonstrate the fold increase of mTagBFP2 levels compared to native *vegfaa* from Gal4/UAS amplification. Error bars represent SEM for *n* = 3 (*vegfaa^{afpUTR}*) or *n* = 4 (*vegfaa^{synCoTC}*) independent biological replicates, consisting of 17 or 18 (*vegfaa^{afpUTR}*) or 14 (*vegfaa^{synCoTC}*) pooled embryos. **g** Vascular labelling by the *Tg(kdrl:EGFP)* reporter demonstrates absence of the dorsal aorta (yellow arrow

in wildtype) in mutants from both CRIMP lines, and branching defects of the trunk intersegmental vessels at 72 hpf. Scale bar 250 µm. **h, i** Both *Ti(vegfaa^{int1}-Gal4vp16/4xnrUAS-mTagBFP2)* (*vegfaa^{afpUTR}*) and *Ti(vegfaa^{int1}-Gal4vp16_synCoTC/4xnrUAS-mTagBFP2)* (*vegfaa^{synCoTC}*) CRIMP *vegfaa* mutants do not undergo genetic compensation. *vegfaa* and *vegfab* mRNA levels in *vegfaa* CRIMP mutants demonstrate complete loss of native *vegfaa* transcript, without a change in *vegfab* (**h**), demonstrating genetic compensation has not been induced. Error bars represent SEM for *n* = 3 (*vegfaa^{afpUTR}*) or *n* = 4 (*vegfaa^{synCoTC}*) independent biological replicates, consisting of 17 or 18 (*vegfaa^{afpUTR}*) or 14 (*vegfaa^{synCoTC}*) pooled embryos. *rpl13* was used as the reference gene. **i** *vegfaa^{afpUTR}* and *vegfaa^{synCoTC}* CRIMP mutants have a reduction in the number of branching intracerebral central arteries (CtAs) similar to the reported non-compensating *vegfaa* promoterless mutant. *vegfaa^{afpUTR-/-}* *n* = 17, *vegfaa^{synCoTC-/-}* *n* = 19, wildtype *n* = 6. Error bars represent SEM. *vegfaa^{afpUTR}* *p* value = 0.0000002, *vegfaa^{synCoTC}* *p* value = 0.0000000003. Source data are provided as a Source Data file. Statistical differences were determined using a two-tailed unpaired *t*-test.

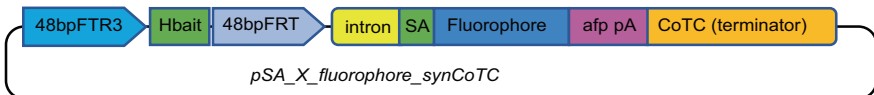

**a   Fluorophore reporter for medium to highly expressed targets**

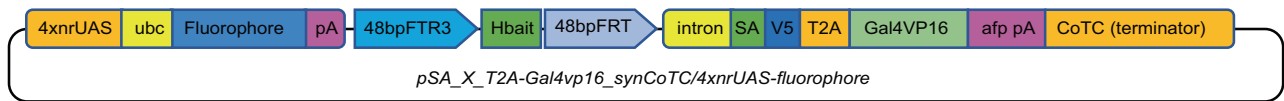

**b   Gal4/UAS-fluorophore reporter for targets with low expression**

**c   Selection of correct CRIMPkit splicing vector**

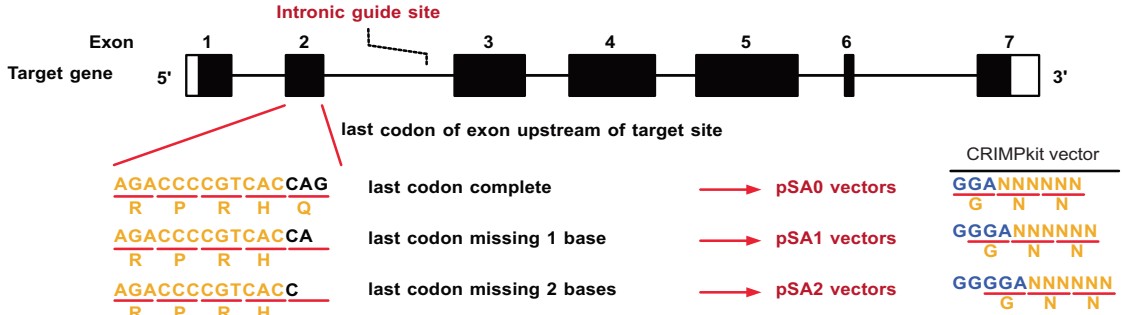

**Fig. 5 | Schematic of CRIMPkit vector design.** CRIMPkit vectors contain a guideRNA target site (Hbait) for CRISPR/Cas9 mediated linearisation. The Hbait site is directly upstream of the 3′ region of *β-actin* intron 2, which contains the splice acceptor for exon-3. One or two bases have been added following the splice acceptor in pSA1 and pSA2 vectors respectively, to ensure the downstream reading frame is preserved. CRIMP vectors with a fluorophore after the splice acceptor **a** are designed for targeting genes with high expression levels. Vectors with Gal4vp16 after the splice acceptor and downstream UAS:fluorophore cassette **b** are for targeting genes with low expression levels. SA splice acceptor, afp pA Ocean pout antifreeze protein 3′UTR until the poly-A sequence. ubc 5′UTR intron of ubiquitin C

CoTC co-transcriptional cleavage. **c** Selection of correct CRIMP splicing vector. The reading frame of the last codon in the exon upstream of the intronic target site determines the correct vector to use in order to preserve the correct reading frame of the downstream coding sequence in a successfully targeted intron. A complete final codon will require a pSA0-vector, a codon missing one base requires a pSA1-vector, and a codon missing two bases requires a pSA2-vector to maintain the correct coding sequence. Black bases indicate the final codon of an exon upstream of an intronic target. Blue bases indicate the sequence incorporated in CRIMP vectors to maintain the reading frame after splicing.

## The CRIMP toolkit (CRIMPkit)

To allow widespread adoption of CRIMP, we developed the CRIMP toolkit, CRIMPkit, consisting of 24 ready-to-use plasmid vectors with the variety of fluorophores and GAL4 amplification described above, and with each vector available in all three possible reading frames. The vector design consists of a highly active guideRNA binding site (Hbait; not present in the zebrafish genome), followed by a splice cassette, and a transgene (outlined in Fig. 5a and the complete list of plasmids in Supplementary Table 3). The splice cassette consists of the 3′-region and splice acceptor from *β-actin* (*actb1*) intron 2 (with three versions of

each plasmid, one in each possible reading frame, Fig. 5c). The splice cassette is followed by the coding sequence of a reporter transgene, and a highly active 3′UTR and poly-A signal derived from ocean pout anti-freeze protein 3′UTR (afpUTR)[27,28]. The CRIMPkit vectors also contain the synCoTC terminator element we generated.

There are multiple reporter transgene options available that consist of either a flexible protein linker (3xGGGGS) followed by a fluorophore (*mTagBFP2, mKate2, mTagBFP2-T2A-splitGFP1-10, or mTagBFP2-T2A-splitGFP11x7*)[18,29,30], or a T2A-Gal4/UAS expression cassette. The 4xnrUAS element used in the Gal4/UAS vectors is

methylation resistant to prevent silencing in subsequent generations[31]. The 4xnrUAS cassettes include a modified *UBC*-intron before either *mTagBFP2* or *mKate2* coding sequences to increase expression levels[27]. *mTagBFP2*[29] and *mKate2*[30] under control of the 4xnrUAS have an additional valine at the second position for increased mRNA stability and expression levels[32].

The Gal4/UAS vectors have a V5 peptide tag (GKPIPNPLLGLDST[33]) included directly after the splice acceptor, before the T2A-Gal4 element, which enables any truncated protein, if produced, to be detected via immunohistochemistry or western blot using an antibody against V5. The *pSA_X_mTagBFP2_synCoTC* vectors also have a flexible linker (GGGGS)-V5 tag at the C-terminal of *mTagBFP2*. All CRIMPkit vectors that contain *mTagBFP2-T2A-splitGFP* have the GGGGS linker-V5 tag on the C-terminal of *mTagBFP2* before the T2A-splitGFP.

In all CRIMPkit vectors the CRISPR/Cas9 guide site is flanked by 48 bp FRT3 and FRT sites, to allow recombination-mediated cassette exchange (RMCE) when induced by FLP recombinase[34]. This exchange is untested but is expected to allow for exchange of the CRIMPkit insertion with alternative sequences if desired.

All CRIMPkit vectors have been codon optimised for zebrafish using the CodonZ software to enhance expression levels[27].

## Discussion

We demonstrate the efficient generation of mutant strains using CRIMP. The CRIMP system provides a significant advantage over current strategies for multiple reasons. Firstly, the CRIMPkit vectors are universal as they are available in all three reading frames after the splice acceptor, and do not require modification depending on the target gene, enabling this toolkit to target any intron of any gene. The kit contains multiple fluorophore reporters (*mTagBFP2*, *mKate2*, and *splitGFP*) and options for both high and low-expressed genes (with and without Gal4/UAS fluorophore amplification). We demonstrate the Gal4/UAS system is able to amplify reporter expression, with transcript levels for the reporter 4−35 times higher in heterozygous carriers than the native transcript in wildtype fish, and the signal is further amplified ~4-fold in homozygous mutants. Whilst we were able to visually detect the reporter gene expression in the *bag3*[mTagBFP2] strain without amplification, we would suggest that GAL4/UAS amplification be used for genes expressed at this level (30 TPM at 24 hpf) or lower to facilitate identification and visualisation. We applied the Gal4 system to increase the levels of fluorescence, but strains generated using these vectors can also be used as driver lines in subsequent experiments.

The protocol is able to achieve a very high proportion of mosaic integrations (57−96%, Table 1 and Supplementary Table 2), improving on previous reports of up to 57%[10]. More significantly, the protocol is able to produce early integration events during the first cell division generating embryos with positive fluorescence throughout one or both halves of the embryonic body plan. Such early integration events have not been reported previously and provide a significant improvement over methods which only generate mosaic integration events after the embryo has reached the 1000-cell stage. In our experience, 100% of such embryos tested successfully transmitted integrations to the next generation. This enables the preselection of founders prior to raising, overcoming the need to raise large clutches of injected embryos, and greatly reducing the time to generate strains, by removing the need to screen for founders, and the number of fish required. Therefore, whilst the frequency of these early integration events is low, the efficiency of the complete process of generating new lines is dramatically improved.

Insertional gene trap approaches are prone to incomplete termination after the transgene. Previous reports have identified the expression of native transcript in 5 out of 9 generated lines, at up to 32% of wildtype transcript levels[35]. This is likely due to poly-A signals being poor transcriptional terminators and allowing read-through of the RNA polymerase II[26,36–38]. Some approaches have attempted to

solve this through the addition of multiple tandem copies of the poly-A sequence[9,39] or a second downstream expression cassette to reduce the level of read-through[9,10]. Whilst it is not clear if a functional protein is produced when termination is not effective, it is possible, due to splicing of the transcript, to exclude the inserted transgene, and some gene trap lines have been described as hypomorphic, demonstrating reduced severity of phenotypes compared to a complete loss of function[35].

We identified reporter expression variation in one of our targeted insertion lines containing Gal4vp16/UAS expression cassette, which we believe was a result of incomplete transcriptional termination interfering with the downstream UAS element, as has been described for other transgenes[40]. Incorporation of a Co-transcriptional Cleavage (CoTC)-type terminator element[26] after the Gal4vp16 poly-A prevented reporter variation and generated more robust reporter expression levels. This is consistent with earlier studies that have shown the ability of CoTC elements to enable complete transcriptional termination[26]. We believe the inclusion of the CoTC element will not only prevent the generation of hypomorphic alleles but may also increase the expression of the reporter due to improved pre-mRNA processing and reduced degradation of the RNA[38].

In mouse knockouts involving exon deletion or replacement with reporters or selection cassettes, alternative transcripts can form, resulting in hypomorphic or gain of function alleles[41]. Similarly, CRISPR/Cas9 and ENU-generated mutations in zebrafish can lead to exon skipping, the use of cryptic splice sites, or alternative start sites[42]. These unpredicted consequences of mutation can greatly complicate the analysis of the mutant lines. The use of the CoTC element would be expected to prevent the formation of alternative downstream gene products. Finally, conventional CRISPR/Cas9 mutagenesis has the potential to generate alleles which undergo genetic compensation triggered by nonsense-mediated decay[1,3]. Insertional mutagenesis provides the benefit of being able to prevent genetic compensation, as nonsense-mediated decay is not activated. While other gene trap style vectors used for insertional mutagenesis are also likely to prevent genetic compensation[9], we have experimentally validated that CRIMPkit vectors generate mutant alleles that do not induce genetic compensation.

Here we provide a protocol and toolkit to facilitate the application of insertional mutagenesis to efficiently generate loss of function alleles. Whilst the provided toolkit is optimised for use in zebrafish, due to codon optimisation of the plasmids, the approach and toolkit could be used in any species.

## Methods

### Ethical statement

Zebrafish (*Danio rerio*) were maintained in the Monash University FishCore facility under breeding colony license MARP/2015/004/BC and ERM22161. The creation of transgenic lines was approved by the School of Biological Sciences Animal Ethics Committee (ERM18912, 21168). All experiments were carried out on embryos of TU/TL background. All animal work was approved by the Monash Animal Ethics Committee in accordance with the care and use of animals for scientific purposes. Fish were anaesthetised for live imaging using Tricaine methanesulfonate (3-amino benzoic acidethylester; Sigma Aldrich, E10521) at a final concentration of 0.016% in E3 embryo medium (5 mM NaCl, 0.17 mM KCl, 0.33 mM CaCl₂, 0.33 mM MgSO₄, 0.00004% [v/v] methylene blue in water, pH 7.2).

### CRISPR/Cas9 insertional mutagenesis

The Alt-R CRISPR/Cas9 injection protocol from Integrated DNA technologies™ that we modified was obtained from: https://sfvideo.blob.core.windows.net/sitefinity/docs/default-source/user-submitted-method/crispr-cas9-rnp-delivery-zebrafish-embryos-j-essnerc46b5a1532796e2eaa53ff00001c1b3c.pdf?sfvrsn=52123407_12.

Intronic guideRNA sites in target genes were identified using the IDT™ online tool (https://sg.idtdna.com/site/order/designtool/index/CRISPR_CUSTOM). Target sites with the highest on-target and off-target scores were selected, with a preference to those located in the 3′ half of the intron. guideRNA target sites used in this study are listed in Supplementary Table 4. All vector sequences are provided in raw data repository detailed below. Integration site sequences are described in Supplementary Fig. 3. A summarised protocol for use in the lab is provided in Supplementary Fig. 4.

### cDNA synthesis and qRT-PCR

Total RNA was extracted using TRIzol® reagent (Invitrogen Life Technologies). cDNA was synthesised from 1 μg of each RNA sample in a 20 μl reaction using ProtoScript® II First Strand cDNA synthesis kit (New England Biosciences) and oligo(dT)20 and random hexamer primers following the supplier's instructions. qRT-PCR was performed on a LightCycler® 480 instrument (v1.5.1.62) using LightCycler® 480 SYBR® Green I Master mix (Roche), as per the manufacturer's protocol. *vegfaa* samples were normalised against *rpl13*[3] and/or *ef1α* as a reference gene. *tdgf1* and *bag3* samples were normalised against the geomean of *rpl13* and *ef1α* as reference genes. Primers for qRT-PCR are listed in Supplementary Table 4. Technical replicates were performed in triplicate for each sample, and each experiment was repeated using at least three independent biological replicates.

### Methylcellulose assay

24 hpf embryos were incubated in 1% methylcellulose for 1 h and subsequently fixed in 4% paraformaldehyde and labelled for Myhc (A4.1025, DSHB, 1:10) and a Goat anti-mouse Alexa Fluor-488 secondary antibody (Thermo Fisher A11001, 1:150) as per ref. [43]. A4.1025 was deposited to the DSHB by Blau, H.M. (DSHB Hybridoma Product A4.1025).

### Imaging

Imaging was performed with an Olympus SZX16 stereomicroscope using pco.camware V04.11/1527/64 bit software. mTagBFP2 and mKate2 fluorescence was visualised using the Chroma™ 49021 and 49008 filter sets, respectively. In order to examine intracerebral central arteries in the brain at 60 hpf, embryos were fixed in 4% paraformaldehyde overnight at 4 °C, washed in PBST (1× PBS, 0.1% Tween 20 (Sigma)), and set in 1% low melting agarose in 0.8 mm fluorinated ethylene propylene (FEP) tubing (Bola, Grünsfeld, Germany). Methylcellulose-treated and labelled fish were mounted on coverslips with ProLong Glass Antifade (Thermo Fisher Scientific P36982). Genotypes were randomised, and the investigator blinded to genotype. Images were taken using a Thorlabs confocal microscope running ThorImageLS v4.0.2019.8191 software (Newton, NJ, USA), with an Olympus 20x water dipping 1.0 NA objective (Tokyo, Japan), pinhole 25 μm, 2.005 μm/pixel, step size 1 μm, averaging 16 frames. Images were analysed using Fiji (ImageJ) software[44].

### Microinjection

Freshly fertilised embryos were injected using a Femtojet microinjector (Eppendorf). Approximately 1 nl of the mix was injected into the cell portion of the embryo at the one-cell stage, usually within the first 15 min of being laid.

### Sequence analysis

Sequence annotation and analysis was performed using the A Plasmid Editor software (ApE)[45].

### Statistics and reproducibility

For gene expression analysis using qRT-PCR, 96 larvae for each condition were collected and genotyped to identify a minimum of 14 homozygous mutants. The number of identified homozygous embryos determined the number of samples for all other genotypes, as indicated in the figure legend. No statistical method was used to pre-determine the sample size. No data were excluded from the analyses. Where mutant fish were compared to WT siblings, samples were randomised, and the investigator was blinded to genotype. Quantification and analysis was performed prior to revealing genotype. qRT-PCR data was analysed using the ΔΔCT method, and statistical differences were determined using a two-tailed unpaired t-test. For Intracerebral central artery analysis, statistical differences were determined using a two-tailed unpaired *t*-test. All statistical analyses were conducted using GraphPad Prism 9 or Microsoft Excel 16.77.

### Reporting summary

Further information on research design is available in the Nature Portfolio Reporting Summary linked to this article.

## Data availability

Source data are provided with this paper. CRIMPkit vectors are available from Addgene as a complete kit #1000000225 [https://www.addgene.org/kits/bryson-richardson-crimpkit/] or individual plasmids: 199469, 199470, 199471, 199472, 199473, 199474, 199475, 199476, 199477, 199478, 199479, 199480, 199481, 199482, 199483, 199484, 199485, 199486, 199487, 199488, 199489, 199490, 199491, 199492, 199493, 199494, 199495, 199496, 199497, 199498. The CRIMPkit vector sequence data has been deposited in GenBank database under accession numbers: PP297139, PP297140, PP297141, PP297142, PP297143, PP297144, PP297145, PP297146, PP297147, PP297148, PP297149, PP297150, PP297151, PP297152, PP297153, PP297154, PP297155, PP297156, PP297157, PP297158, PP297159, PP297160, PP297161, PP297162. The raw data and statistical analysis files used in this study are available in the Monash University Bridges data repository https://doi.org/10.26180/c.7217754.v5[46] Source data are provided with this paper.

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

## Acknowledgements

The authors would like to thank Dr Harold Burgess for supplying the CodonZ software. This work was supported by research grants awarded to RBR from the University of Pennsylvania Orphan Disease Center in partnership with Cure CMD (MDBR-20-110-CMD), from AFM-Telethon (23703), and funding from the National Health and Medical Research Council (ID 1146321). The contents of this work is solely the responsibility of the authors and do not reflect the views of NHMRC.

## Author contributions

Conceptualisation: L.B.M. and R.J.B.-R. Experiments and analyses: L.B.M., V.C., S.O., R.J.S., C.S., O.M., and C.L. Supervision: R.B.R. Writing: L.M. and R.J.B.-R. All authors reviewed and approved the manuscript.

## Competing interests

The authors declare no competing interests.
