## [Peer Review File · Nature Communications]

CRIMP: a CRISPR/Cas9 Insertional Mutagenesis Protocol and ToolkitEditorial Note: Parts of this Peer Review File have been redacted as indicated to remove third-party material where no permission to publish could be obtained.

Reviewer #1 (Remarks to the Author):

Reviewer comments for the authors:

In this manuscript, Miles et al. describe a novel, targeted insertional mutagenesis system, CRIMP, as well as an associated plasmid toolkit, CRIMPkit, containing 30 plasmids. Their strategy is aimed to disrupt native gene expression by inducing complete transcriptional termination to produce null mutant alleles without inducing genetic compensation.

Major comments overall:

Overall, the manuscript presents a highly valuable and detailed resource for gene editing strategies using CRISPR-Cas9. The authors clearly discuss this in the last paragraph at the end of the manuscript, with emphasis that the induction of genetic compensation can be avoided. Further, the CRIMPkit plasmids are already deposited with addgene, for easy and fast access from almost everywhere. The biggest limitation is that this method was only tested in 3 loci: *tdgf1*, *vegfaa*, and *actc1b*, which of course is a limitation to every new method that is being developed.

To make the manuscript more accessible to the readership, the authors are encouraged to exactly document how many embryos were injected with which plasmids and how many adults were screened for germline transmission in addition to how many adults inserted the transgene (between 1 and 3?). By incorporating this information, readers will get an idea of how efficient this method is, if it varies between loci, and how to apply this method in their own labs for future experiments. Questions below are about the *tdgf1* locus, but the authors are encouraged to do the same for *vegfaa* and *actc1b*.

- Page 2, last paragraph, "95% of injected embryos had detectable mosaic mTagBFP2 expression,..." The authors are encouraged to include how many embryos they injected. In the reporting summary, they mention 96 – is this the case for all injections?

- Page 2, last paragraph, "...and included one embryo displaying the correct expression pattern throughout the ventral half of the body plan (0.7% of injected embryos)" – is there a figure panel associated with it?

- Page 2, last paragraph, "...When raised to adulthood and crossed to wildtype this individual passed on the transgenic insertion,"- how many embryos in a clutch had the transgenic insertion?

Minor comments:

- The authors mention point 5 in their supplemental Fig. S2: "Collect embryos within 5 minutes of being laid" and "In our experience injecting into the embryos within the first 15 minutes increases the likelihood of integration before (or during) the first cell division,". This needs! to get more visibility, and the authors are encouraged to move this into the main text, if applicable.

- As a suggestion, that probably crossed the authors mind as well, could the vector names in Table S1 get an abbreviation? It might make it easier for readers to read through the text.

Reviewer #2 (Remarks to the Author):

Miles et al. generated a collection of 30 plasmids called CRIMPkit to be used as universal donors for targeted insertion into an intron to achieve a gene-trap mutagenesis effect through CRISPR/Cas9 system. These plasmids set covered all three possible reading frames, and therefore could be directly used for targeting any gene with any reading frame. In addition, a fluorescent reporter is included in each plasmid, enabling labeling of the mutant allele for easy identification. Gal4/UAS system was

employed in some plasmids to enhance the expression of the fluorescent reporter gene. SplitGFP was also introduced as an alternative design for direct identification of "homozygous" mutants (compound heterozygotes). Based on this toolkit, the authors designed a protocol called CRIMP to achieve high-efficiency targeted insertional mutagenesis which theoretically could bypass genetic compensation, and demonstrated its feasibility in zebrafish in three target genes. Finally, the authors emphasized several advantages of their system, including the ready-to-use and universality of their donor vectors, feasibility for both high and low expression target genes, availability of different fluorescent reporters, high efficiency of integration events, and avoidance of genetic compensation. The data they presented are impressive and seems reliable; however, the methodology on its own is rather superficial and lacks essential and important information and comparisons for the evaluation of the significance and fundamental contributions of this manuscript. Detailed concerns as the following:

1. About the CRIMP protocol: No sufficient introduction on the CRIMP protocol, which is their essential improvement and should have been emphasized and described in details. How special different is their protocol comparing with current publications? Any special steps and tricks in order to achieve highly efficient targeted integration? The authors should provide detailed description on their protocol and also provide enough experimental results showing the comparison between their method and current methods from others, to demonstrate the advantages and significance of their new approach.

2. About high efficiency of integration: (1) Are there any particular improvements in methodology accounting for the highly-efficient insertion events? The high-efficiency might be due to high indel efficiency and easy accessibility of the target sites per se rather than the improvement of the method itself. Any preselection or consideration for choosing the intron target site? Since there is no comparison in efficiencies with currently available methods, it's impossible to distinguish or rule out these possibilities. (2) In each target site, very limited amount of founder embryos (only particular ones, i.e., those displaying the highest level of mosaicism or showing half-body expression pattern) were screened for germline transmission of the insertion events. Obtaining these special embryos seems rather tricky, how were the other majority founder embryos? How efficient were they in germline transmission? These data should be more informative and useful than only displaying the embryos derived from rare integration events. Accordingly, why there were no such particular embryos with the CoTC-containing vector targeting vegfaa gene (considering there were even more embryos injected in the CoTC group. Fig. S5: 398 vs 180 injected embryos)?

3. About avoiding genetic compensation: In fact, this is a common feature for targeted insertion capable of inducing gene-trap effect rather than a special advantage of the method described in this manuscript. Essentially all the published methods for targeted gene-trap through NHEJ-mediated intron insertion could bypass genetic compensation (e.g., Li, et al., 2019, eLife; Han, et al., 2021, Protein Cell). Thus, the authors' statement that their system has advantages over current strategies because it can cause gene knockout "without the induction of genetic compensation" is inappropriate.

4. About the plasmids set: The donor vectors compatible with all three reading frame and could be directly used is plausible. But the necessity of the other addition of the components were not sufficiently demonstrated and discussed. For example, (1) why Gal4/UAS should be included? When this enhancement should be used? How different with and without Gal4/UAS amplification of the fluorescent reporter signal for the same target site? How helpful is the Gal4/UAS system in enhancing the fluorescent signal? The authors should test some low expression genes, such as flk (Li, et al., 2015, Cell Research; PMID: 25849248), and compare the results with and without Gal4/UAS. (2) How useful are the 5'-UTR targeting Kozak vectors? The authors listed these vectors but did not mention any results derived from them.

5. About the target genes: All the three target genes tested here have a broad expression pattern, which is much easier in identification of fluorescent founder embryos and provides a better chance to reveal successful integration. How about late expression and/or narrow expression pattern genes? The authors should provide corresponding data to solve this issue or at least discuss this issue.

6. Minor issue: Need to provide a bright field image showing the structure the embryo in Fig. S3, to help for better illustration and understanding of the result.

In summary, the core design of the donor vector for targeted integration is not new, just a set of gene-trap vectors covering three different reading frames. Therefore, the vector collection on its own could not demonstrate sufficient novelty of this work. The employment of the Co-Transcriptional Cleavage (CoTC) type terminator element seems to be the only notable improvement in the vector design. In contrast, the experimental protocol could be an important improvement on the originality; however, this part was barely described in the manuscript, preventing from drawing any conclusion regarding this point. Overall, in its current form, the manuscript doesn't seem to have sufficiently significant methodology improvements, especially with limited successful examples (only three target genes). These issues make the manuscript insufficient for publication in NC as submitted.

We thank the reviewers for their suggestions which have significantly improved the manuscript. We have expanded the manuscript, including changing the format from a brief communication to an article, to allow all of the information requested to be presented.

Reviewer #1 (Remarks to the Author):

Reviewer comments for the authors:

In this manuscript, Miles et al. describe a novel, targeted insertional mutagenesis system, CRIMP, as well as an associated plasmid toolkit, CRIMPkit, containing 30 plasmids. Their strategy is aimed to disrupt native gene expression by inducing complete transcriptional termination to produce null mutant alleles without inducing genetic compensation.

Major comments overall:

*Overall, the manuscript presents a highly valuable and detailed resource for gene editing strategies using CRISPR-Cas9. The authors clearly discuss this in the last paragraph at the end of the manuscript, with emphasis that the induction of genetic compensation can be avoided. Further, the CRIMPkit plasmids are already deposited with addgene, for easy and fast access from almost everywhere. The biggest limitation is that this method was only tested in 3 loci: *tdgf1*, *vegfaa*, and *actc1b*, which of course is a limitation to every new method that is being developed.*

We have added two additional lines to the manuscript, including another new gene, *bag3*. As a result, there are now four genes and five different target sites, for which we have applied the approach.

*To make the manuscript more accessible to the readership, the authors are encouraged to exactly document how many embryos were injected with which plasmids and how many adults were screened for germline transmission in addition to how many adults inserted the transgene (between 1 and 3?). By incorporating this information, readers will get an idea of how efficient this method is, if it varies between loci, and how to apply this method in their own labs for future experiments. Questions below are about the *tdgf1* locus, but the authors are encouraged to do the same for *vegfaa* and *actc1b*.*

This data has been removed from the supplementary data and included in a table in the main manuscript. The table details the number of embryos injected and screened, for all lines included in the manuscript.

Table 1. Integration efficiency

Target gene (protocol optimisation)	Injection efficiency				Founder screening			
	Injected	Mosaic	Half body plan	Total positive		Raised	Screened	Positive
actc1b-mTagBFP2-afpUTR	148	20 (13.4%)	2 (1.3%)	22 (15%)	Mosaic	22	20	0 (0%)
					Half body plan	2	1	1 (100%)
					Total	24		
actc1b-mTagBFP2-sGFP1-10	93	12 (12.9%)	2 (2.2%)	14 (15.1%)	Mosaic	2	2	0 (0%)
					Half body plan	1	1	1 (100%)
					Total	3		
actc1b-mTagBFP2-sGFP11x7	377	42 (11%)	1 (0.2%)	43 (11.4%)	Mosaic	5	4	1 (25%)
					Half body plan	1	1	1 (100%)
					Total	11		
								
(Freshly prepared guideRNA, +>30 min incubation at 37°C)								
bag3-mTagBFP2-afpUTR	120	90 (75%)	-	90 (75%)	Mosaic	4	4	1 (25%)
					Half body plan	-	-	-
					Total	4		
tdgf1-Gal4-afpUTR	136	128 (94.1%)	1 (0.7%)	129 (94.9%)	Mosaic	8	0	0
					Half body plan	1	1	1 (100%)
					Total	9		
vegfaa-Gal4-afpUTR	180	172 (95.6%)	1 (0.6%)	173 (96.1%)	Mosaic	11	1	0
					Half body plan	1	1	1 (100%)
					Total	12		
vegfaa-Gal4-synCoTC	398	378 (95%)	-	378 (95%)	Mosaic	11	6	3 (50%)
					Half body plan	-	-	-
					Total	11		

- Page 2, last paragraph, “95% of injected embryos had detectable mosaic mTagBFP2 expression,…” The authors are encouraged to include how many embryos they injected. In the reporting summary, they mention 96 – is this the case for all injections?

We have included this information in Table 1, and discuss this in the main text for each insertion line.

- Page 2, last paragraph, “...and included one embryo displaying the correct expression pattern throughout the ventral half of the body plan (0.7% of injected embryos)” – is there a figure panel associated with it?

We have moved this image from supplementary data into the new Fig1, as panel E.

- Page 2, last paragraph, “...When raised to adulthood and crossed to wildtype this individual passed on the transgenic insertion,”- how many embryos in a clutch had the transgenic insertion?

We have included in Table 1 the number of fish screened and the number identified as transmitting the insertion to the next generation.

Minor comments:

- The authors mention point 5 in their supplemental Fig. S2: "Collect embryos within 5 minutes of being laid" and In our experience injecting into the embryos within the first 15 minutes increases the likelihood of integration before (or during) the first cell division,". This needs! to get more visibility, and the authors are encouraged to move this into the main text, if applicable.

To address this comment, and those of Reviewer 2, we have added a Protocol optimisation section into the manuscript text. This section outlines the protocol design, and includes the following:

"In order to facilitate early integration events during this first cell division, we used pre-complexed guideRNA/Cas9 ribonucleoprotein (RNP) and targeting plasmid in our injection mix. We aimed to collect embryos within 5 minutes of fertilisation and perform injections within 15 minutes to ensure the highest likelihood for early integration events during this first cell division."

- As a suggestion, that probably crossed the authors mind as well, could the vector names in Table S1 get an abbreviation? It might make it easier for readers to read through the text.

Thank you for the suggestion, we have included an abbreviated name column in Table S1 which we hope will make it easier for the readers.

Reviewer #2 (Remarks to the Author):

Miles et al. generated a collection of 30 plasmids called CRIMPkit to be used as universal donors for targeted insertion into an intron to achieve a gene-trap mutagenesis effect through CRISPR/Cas9 system. These plasmids set covered all three possible reading frames, and therefore could be directly used for targeting any gene with any reading frame. In addition, a fluorescent reporter is included in each plasmid, enabling labeling of the mutant allele for easy identification. Gal4/UAS system was employed in some plasmids to enhance the expression of the fluorescent reporter gene. SplitGFP was also introduced as an alternative design for direct identification of "homozygous" mutants (compound heterozygotes). Based on this toolkit, the authors designed a protocol called CRIMP to achieve high-efficiency targeted insertional mutagenesis which theoretically could bypass genetic compensation, and demonstrated its feasibility in zebrafish in three target genes. Finally, the authors emphasized several advantages of their system, including the ready-to-use and universality of their donor vectors, feasibility for both high and low expression target genes, availability of different fluorescent reporters, high efficiency of integration events, and avoidance of genetic compensation. The data they presented are impressive and seems reliable; however, the methodology on its own

is rather superficial and lacks essential and important information and comparisons for the evaluation of the significance and fundamental contributions of this manuscript. Detailed concerns as the following:

1. About the CRIMP protocol: No sufficient introduction on the CRIMP protocol, which is their essential improvement and should have been emphasized and described in details. How special different is their protocol comparing with current publications? Any special steps and tricks in order to achieve highly efficient targeted integration? The authors should provide detailed description on their protocol and also provide enough experimental results showing the comparison between their method and current methods from others, to demonstrate the advantages and significance of their new approach.

The original submission was intended as a brief communication. To address these comments, we have expanded the manuscript text to include the sections detailing the CRIMPkit vector design (previously in Methods), and details of the protocol design and additional optimisation steps that were undertaken when generating the initial lines.

As well as providing an efficiency table, we have compared the frequency of integration events (75-96% using CRIMP) to the highest previous reported frequency we are aware of (57.1%, Han et al, 2020) in the discussion. However, we have also highlighted in the discussion that we believe that the early integration events, not previously reported, is the more significant advance and have improved the description of this in the discussion, emphasising the efficiency of line generation as a whole, rather than the frequency of integration events alone.

2. About high efficiency of integration: (1) Are there any particular improvements in methodology accounting for the highly-efficient insertion events? The high-efficiency might be due to high indel efficiency and easy accessibility of the target sites per se rather than the improvement of the method itself. Any preselection or consideration for choosing the intron target site? Since there is no comparison in efficiencies with currently available methods, it's impossible to distinguish or rule out these possibilities.

We have added additional details of the development of the protocol to the main text, including the changes to previous approaches that account for the high-efficiency of mosaic integration events and the novel half-body plan positive embryos.

We have also included the additional details of target site selection in the methods section.

“Intronic guideRNA sites in target genes were identified using the IDT™ online tool (https://sg.idtdna.com/site/order/designtool/index/CRISPR_CUSTOM) Target sites with the highest on-target and off-target scores were selected, with a preference to those located in the 3' half of the intron. guideRNA target sites used in this study are listed in Table S2.”

In the main text and discussion, we have included the comparison of CRIMPKit integration efficiencies with the highest efficiency reported in previous approaches.

(2) In each target site, very limited amount of founder embryos (only particular ones, i.e., those displaying the highest level of mosaicism or showing half-body expression pattern) were screened for germline transmission of the insertion events. Obtaining these special embryos seems rather tricky, how were the other majority founder embryos? How efficient were they in germline transmission?

Given that half of the entire body plan of these embryos expresses the reporter we find identifying these embryos straightforward and, despite them being rare, this is one of the major benefits of the protocol, as there is no need to screen additional fish as these rare early events always lead to germline transmission of the integration.

Unfortunately, we did not test all of the injected fish for transmission, however, for the limited number of such fish that we did screen transmission was ~25% and we have provided a breakdown of the numbers for each line in Table 1.

These data should be more informative and useful than only displaying the embryos derived from rare integration events.

We have expanded in the discussion the benefit of identifying these rare integration events. We believe these rare events are incredibly useful. Identifying a single such fish just 24 hours after injection is sufficient for us to know we have successfully generated the strain. We can reduce the numbers of animals used by raising very small numbers of fish, and greatly reduced the effort and time required to establish the line as we do not need to test lots of potential founders for germline transmission. We believe many users will inject until they identify and raise the fish arising from these early integrations, rather than the mosaic expressing fish.

We have however, included data on the frequency of mosaicism and germline transmission, which also improves on previously reported approaches.

Accordingly, why there were no such particular embryos with the CoTC-containing vector targeting vegfaa gene (considering there were even more embryos injected in the CoTC group. Fig. S5: 398 vs 180 injected embryos)?

The following sentence has been added to the text to address the absence of half-body positive embryos in the vegfaa_synCoTC injection:

“We did not detect any embryos with complete expression in one half of the entire body plan in this injection, however, this might have been due to a delay in injection timing, potentially missing the time window required for early integration during the G2 phase of the first cell division.”

3. About avoiding genetic compensation: In fact, this is a common feature for targeted insertion capable of inducing gene-trap effect rather than a special advantage of the method described in this manuscript. Essentially all the published methods for targeted gene-trap through NHEJ-mediated intron insertion could

bypass genetic compensation (e.g., Li, et al., 2019, eLife; Han, et al., 2021, Protein Cell). Thus, the authors' statement that their system has advantages over current strategies because it can cause gene knockout "without the induction of genetic compensation" is inappropriate.

We apologise for the poor phrasing, we meant that compared to the more commonly used CRISPR/Cas9 generation of indels, this approach has the advantage of preventing genetic compensation. We did not intend to claim this was an advantage of CRIMP over other insertional approaches. We have modified the text in the introduction to make this clearer:

“The introduction of nonsense mutations from the insertion of stop cassettes or generation of indel mutations has the potential to induce genetic compensation¹⁻³, masking loss of function phenotypes.

And in the discussion:

“Finally, conventional CRISPR/Cas9 mutagenesis has the potential to generate alleles which undergo genetic compensation triggered by nonsense mediated decay^{1,3}. Insertional mutagenesis provides the benefit of being able to prevent genetic compensation, as nonsense mediated decay is not activated. While other gene-trap style vectors used for insertional mutagenesis are also likely to prevent genetic compensation⁹, we have experimentally validated that CRIMPkit vectors generate mutant alleles that do not induce genetic compensation”

4. About the plasmids set: The donor vectors compatible with all three reading frame and could be directly used is plausible. But the necessity of the other addition of the components were not sufficiently demonstrated and discussed. For example, (1) why Gal4/UAS should be included? When this enhancement should be used?

We have added additional details to the manuscript on the expression levels of the genes for which amplification of the report signal was necessary, and a new line for the gene *bag3*, which was the lowest expressed gene we have analysed to date that was visible without amplification. These lines, together with the information available on their expression levels from RNAseq datasets, provides a guide for when amplification should be used.

We have also added this in the discussion:

“Whilst we were able to visually detect the reporter gene expression in the *bag3^{mTagBFP2}* strain without amplification, we would suggest that GAL4/UAS amplification be used for genes expressed at this level (30 TPM at 24 hpf) or lower to facilitate identification and visualisation.”

How different with and without Gal4/UAS amplification of the fluorescent reporter signal for the same target site? How helpful is the Gal4/UAS system in enhancing the fluorescent signal?

We were unable to visualise the unamplified signal for *tdgf1* or *vegfaa* but were able to observe the reporter following amplification providing an indication of the improvement in signal strength and have added this information to the text. To further measure the amplification levels for the Gal4/UAS lines and describe their application, we have added in qRT-PCR data for the strains comparing the expression of the reporter to the endogenous gene. All Gal4/UAS lines showed a significant increase in expression compared to the endogenous gene with amplification ranging from 4-133 fold.

We have presented the qRT-PCR results for each amplified strain and added to the discussion.

“We demonstrate the Gal4/UAS system is able to amplify reporter expression, with transcript levels for the reporter 4-35 times higher in heterozygous carriers than the native transcript in wildtype fish, and the signal is further amplified ~4-fold in homozygous mutants. Whilst we were able to visually detect the reporter gene expression in the *bag3^{mTagBFP2}* strain without amplification, we would suggest that GAL4/UAS amplification be used for genes expressed at this level (30 TPM at 24 hpf) or lower to facilitate identification and visualisation. We applied the Gal4 system to increase the levels of fluorescence, but strains generated using these vectors can also be used as driver lines in subsequent experiments.”

The authors should test some low expression genes, such as flk (Li, et al., 2015, Cell Research; PMID: 25849248), and compare the results with and without Gal4/UAS.

The native expression levels in transcripts per million (TPM) for all target genes has been included in the text, this data was obtained from the expression atlas (White., et al 2017).

Reviewer 2's suggests adding some low expression genes such as *flk (kdr)* which is expressed at 11 transcripts per million (TPM) at 24 hpf. However, our two current examples of genes with low expression, *vegfaa* and *tdgf1*, have expression levels of 9 and 10 TPM respectively, which is at similar or lower levels than the suggested *flk (kdr)* target. Therefore, we believe that the additional information on the expression levels of these genes demonstrates the benefit of amplification.

Reviewer 2's also suggests comparing the results with and without Gal4/UAS on a target with low expression levels. The use of non-amplifying targeting vectors for these genes with low expression levels, *vegfaa* and *tdgf1*, did not result in any detectable level of fluorescent reporter expression. We have included the description of this in the text as follows

“Targeting of a splice acceptor-*mTagBFP2* vector into intron-3 of *tdgf1* did not result in any detectable *mTagBFP2* fluorescence in the injected embryos, which we suspected was due to the low expression levels of the *tdgf1* gene.”

“Similar to the results for *tdgf1*, targeting of intron-1 of *vegfaa* with a splice acceptor-*mTagBFP2* vector did not result in any detectable *mTagBFP2* fluorescence in the injected embryos”

(2) How useful are the 5'-UTR targeting Kozak vectors? The authors listed these vectors but did not mention any results derived from them.

We believe these would be useful for a subset of genes and so included them in the kit but as we have not verified these, we have therefore removed them from the article.

5. About the target genes: All the three target genes tested here have a broad expression pattern, which is much easier in identification of fluorescent founder embryos and provides a better chance to reveal successful integration. How about late expression and/or narrow expression pattern genes? The authors should provide corresponding data to solve this issue or at least discuss this issue.

The target genes *tdgf1* and *vegfaa* have restricted and low level expression, and we were able to identify these strains successfully. We have clarified in the text that these gene have low levels of expression (providing the transcripts per million data) and restricted expression, so that this is clear to the reader.

For genes expressing in an even smaller set of cells, the challenge is in the imaging, and not in the mutagenesis, however, the amplification approach detail in the manuscript will facilitate the imaging. There is no reason why the approach could not be used for later expressing genes, we screened at early stages as this was sufficient for our examples and early development is the timepoint at which the majority of zebrafish research is conducted.

6. Minor issue: Need to provide a bright field image showing the structure the embryo in Fig. S3, to help for better illustration and understanding of the result.

To make this image more understandable we have included a dotted line outlining the structure of the embryo, as well as one indicating the embryos body axis. Unfortunately, we did not have the corresponding image captured in brightfield.

In summary, the core design of the donor vector for targeted integration is not new, just a set of gene-trap vectors covering three different reading frames. Therefore, the vector collection on its own could not demonstrate sufficient novelty of this work. The employment of the Co-Transcriptional Cleavage (CoTC) type terminator element seems to be the only notable improvement in the vector design. In contrast, the experimental protocol could be an important improvement on the originality; however, this part was barely described in the manuscript, preventing from drawing any conclusion regarding this point. Overall, in its current form, the manuscript doesn't seem to have sufficiently significant methodology improvements, especially with limited successful examples (only three target genes). These issues make the manuscript insufficient for publication in NC as submitted.

We have revised the text of the manuscript as requested to expand on the protocol improvements that have led to higher levels of fish with integration events than previously reported and in the early integration of the targeting vectors, both of which are significant improvements on previous approaches. The increased frequency of integration events will improve the efficiency with which new strains can be generated, but the early integration events in particular allows the identification of fish, very soon after injection, that will definitely pass on the integrated reporter to the next generation. This removes uncertainty in the generation of new strains, eliminates the time required to screen potential founders, and reduces the number of animals that need to be raised. Together these improves greatly improve the generation of new strains.

We have included an additional line targeting *bag3* that we had previously generated and not reported but were unable to generate further additional strains in the time provided. We believe the four genes and seven associated lines we have reported is sufficient to demonstrate the success of the approach.

We do not claim that the provision of a kit is novel, it is, as stated by the reviewer, a series of gene trap vectors. However, previous insertional mutagenesis approaches have not been widely adopted. We believe that the design of the vectors incorporating many useful features to facilitate the generation of mutant strains, the inclusion of a transcriptional terminator, together with the universal nature of the provided kit, allowing targeting of any gene without customisation will result in its widespread adoption by other researchers and have a significant impact on the research community.

Reviewer #1 (Remarks to the Author):

The authors have addressed all raised points in their response, with some rearrangements to the text, figures and supplemental data. The revised manuscript is sound, with only minor edits:

- Figure 1: panel a, gene name *actc1b* in italics; panel b, intron or HiFi-Cas9 not in italics, double check.
- Include table legend for Table 1.
- Figure S1: double check spelling, e.g. "targetting", include figure legend.
- Include table legend for Table S2.
- All figures and tables look great, nice color choices!

Reviewer #2 (Remarks to the Author):

The revised manuscript showed certain improvements. The plasmid toolkit provided here is certainly valuable, and the split-GFP and Gal4/UAS modifications are a beneficial improvement for gene tagging systems. Nevertheless, I think the mutagenesis protocol with easy identification of germline transmission is the major point of originality of this work. The authors now provided more details on how to achieve efficient insertional mutations by using their method, including early injection and Cas9/gRNA incubation time. However, much of the description of specific steps of the protocol is based on speculations without support from proper comparison of experiments. For example:

- (1) The entire paragraph from lines 67 to 78 is speculative, there is no data supporting their claim of G2 phase targeting. This is of course an interesting idea worthy of further study, but the claims linking knockin injection timing to a specific cellular and molecular mechanisms of knockin are inappropriate and cannot be supported by data presented in the current manuscript. Additional claims were made based explicitly on speculation about timing of knockin, though interesting and reasonable guesses of what caused observed results, are again not supported by proper data (such as line 108 to 112).
- (2) Line 134 should not claim significance without statistical tests. Furthermore, the claim that gRNA incubation time improves activity (lines 130 to 136) cannot be compared across different CRISPR target sites, as every CRISPR paper since 2012 has shown that different spacers have different inherent activity. The 30 minute incubation suggested here doesn't match the incubation time in the supplemental protocol (Figure S1 step 4). If 30 minute incubation is apparently superior to shorter incubation, why was the suggested 1 hour incubation not tested? Furthermore, the authors should mention the cost-benefit of longer incubation with potential in vitro cleavage of the plasmid donor, as injection of linearized donor is generally not suitable for gene knockin.
- (3) The authors emphasize that the contribution of this manuscript is the half-body fluorescence allowing for easier pre-screening of founder fish. In two of the seven lines tested, no half-body founders were observed. The authors should discuss what approaches are necessary to achieve these founders, and explain whether the method can generally be expected to work only 5/7 of the time.

Additional comments:

- (1) One can easily overcome genetic compensation by inhibiting *upf3a* expression or disrupting *upf3a* gene when generating indel mutations of their target genes, and screening for indel mutations is much easier than screening for insertional mutations. The authors should also discuss the disadvantages of their method in preventing genetic compensation and why researchers should choose their method to avoid genetic compensation.
- (2) The authors note several embryos showing fluorescence in half of the body (line 157, 173). The only comparable micrograph shown (Fig 1E) is captioned as having fluorescence in "one half of the ventral body plan" (line 550). Half of the ventral body suggests one quarter of the whole body. Are the embryos not fluorescent if observed dorsally? The authors need to make clear what they observed.
- (3) Please provide the data of germline mosaicism of each positive founder fish in Table 1.
- (4) Any phenotype of the *actc1b* splitGFP mutant embryos?
- (5) Does *Ti(tdgf1int3-Gal4vp16/4xnrUAS-mTagBFP2)* also show variable expression levels of *mTagBFP2*?
- (6) The Abstract is poorly written, did not emphasize the particular improvements in their

methodology.

(7) The manuscript is poorly organized, no Results section was found and the results were mixed with Introduction.

Considering the above concerns, I think the authors should properly emphasize the improvements on the mutagenesis methods by providing sufficient experimental evidence, besides introducing the design and generation of universal plasmids set. Otherwise the innovation of the manuscript is quite limited and it's more appropriate for this manuscript to publish as a resource paper (design and collections of the universal plasmids set) in a more specialized journal rather than NC.

We thank the reviewers for the comments and have made additional changes to the manuscript which we believe has strengthened it further, providing additional experimental evidence to illustrate the advantages of the protocol and approach described. We respond to each of the reviewers comments in detail below.

REVIEWER COMMENTS

Reviewer #1 (Remarks to the Author):

The authors have addressed all raised points in their response, with some rearrangements to the text, figures and supplemental data. The revised manuscript is sound, with only minor edits:

- Figure 1: panel a, gene name *actc1b* in italics; panel b, intron or HiFi-Cas9 not in italics, double check.

We have corrected these errors in Figure 1, the updated figure is below.

- Include table legend for Table 1.

We have added a legend for Table 1 as below

“Table 1. Integration efficiency, founder screening, and germline transmission rates. The integration efficiency of each target site in F₀ embryos is located on the left side of the table. The founder screening and germline transmission rates of successful founders are located on the right side of the table. Targeting events below the dashed line are utilising the optimised protocol of freshly complexed guideRNA and an extended incubation time at 37 °C prior to injection.”

- Figure S1: double check spelling, e.g. “targetting”, include figure legend.

We have changed the spelling to ‘targeting’ and checked the remaining text for errors. We have added a brief legend for the protocol which is now Figure S4.

“Figure S4: CRIMP Injection Protocol. A summary of the protocol for lab use.”

- Include table legend for Table S2.

We have added a legend to the table, now Table S4, as shown below.

Genotyping primers	Primer sequence		
tdgf1_int3_F	TCTCTGGGAATGTCATGGCT		
tdgf1_int3_R	AGGTGTCTAGCATTGCCGTA		
β-intron_R	ACCAGCTCACCGAGAAATGA		
mTagBFP_end_F	AGCTGGGACACAAGCTGAAT		
vegfaa_int2_F2	TCTGTAGGCGGAAAGAAGA		
vegfaa_int2_R4	AGCCAAATATTGTCCTAACAAAGTG		
T2A_gal4_qPCR_R	GAAGTTTCATTGGGCCAGGG		
bag3_int3_F	TGAATGAGCAAGGGTTTCTGC		
bag3_ex3_R	GGGCACTTGTTCACGGTAGA		
			
qRT-PCR primers			
vegfaa_qPCR_F	CCCACGATATCACACTCGGT		
vegfaa_qPCR_R	GGATGTACGTGTGCTCGATCT		
vegfab_qPCR_F	GGTGCTGCAATGATGAAATG		
vegfab_qPCR_R	TGTCACCCTGATGACGAAGA		
rpl13_qPCR_F	TAAGGACGGAGTGAACAACCA		
rpl13_qPCR_R	CTTACGTCTGCGGATCTTTCTG		
EF1α_F	CTGGAGGCCAGCTCAAACAT		
EF1α_R	ATCAAGAAGAGTAGTACCGCTAGCATTAC		
mTagBFP2_R	GGTTGTCCACAGTCCCTCC		
V5_R	GGGTTTGGGATTGGCTTTCC		
UBC_F	AGCTCCTCCACACGAATTC		
			
crRNA guide sequence(PAM)		On-target score	Off-target score
tdgf1_int3	TGTTACATTGGTTGATAATA(GGG)	73	42
vegfaa_int1	ATACTAATATAGGACCACCC(AGG)	91	94
actc1b_int2	TAGATTTAGGACAAGTCTGCG(TGG)	87	79
actc1b_int4	ATCTACTTAGACCTCACATG(TGG)	79	92
bag3_int2	TCCTCACACAAAAGAGCCTG(CGG)	63	78
Hbait	GAAGATCGGCCACTACATTC(TGG)		

Table S4. Primer and guideRNA sequences. Genotyping primers, qRT-PCR primers, and guide site sequences using in this study. GuideRNA PAM sequence is indicated by brackets. IDT predicted on-target and off-target scores are provided for each guide site.

- All figures and tables look great, nice color choices!

Reviewer #2 (Remarks to the Author):

The revised manuscript showed certain improvements. The plasmid toolkit provided here is certainly valuable, and the split-GFP and Gal4/UAS modifications are a beneficial improvement for gene tagging systems. Nevertheless, I think the mutagenesis protocol with easy identification of germline transmission is the major point of originality of this work. The authors now provided more details on how to achieve efficient insertional mutations by using their method, including early injection and Cas9/gRNA incubation time. However, much of the description of specific steps of the protocol is based on speculations without support from proper comparison of experiments. For example

(1) The entire paragraph from lines 67 to 78 is speculative, there is no data supporting their claim of G2 phase targeting. This is of course an interesting idea worthy of further study, but the claims linking knockin injection timing to a specific cellular and molecular mechanisms of knockin are inappropriate and cannot be supported by data presented in the current manuscript.

The description of the protocol steps is not in any way speculative, it describes the exact steps we undertook in generating the strains presented.

The referenced text in lines 67 to 78 was referring to previous data published in the literature by Gutierrez-Triana et al., and Bouldin and Kimelman, a research article and review that we referenced alongside this background information. We believe the data presented in these references was of high quality and the evidence presented and discussed in these references supports the arguments they present. We do not investigate the mechanism of integration and make no claims that we have and therefore, we agree that we have no data to support the mechanism.

We have now rephrased the text in this section, included alternative references, and hopefully made it clearer that we are referring to published work and not our own data, replacing the previous text with

“Zebrafish have a pause phase in the cell-cycle during the first cell division¹⁴, which results in an extended cell cycle length during this first cleavage event¹⁵, after which they undergo extremely rapid cell divisions, which continues until the midblastula-transition where G2 cell cycle pauses are acquired¹⁶ and the zygotic genome becomes active.”

Additional claims were made based explicitly on speculation about timing of knockin, though interesting and reasonable guesses of what caused observed results, are again not supported by proper data (such as line 108 to 112).

This text was included as the reviewer previously asked why we did not see early integration events in two of the strains generated and so we added text to propose a reason for this. The referred to text begins with 'We suspect' and then later says "which may have resulted in a large proportion being injected after the optimal time window for integration during the G2 phase of the first cell division". We thought this was sufficient caution in the presentation of a reasonable hypothesis to explain our results and did not make any claim that we had demonstrated this to be the case. However, we have now further modified this text to remove reference to the G2 phase.

"We suspect that the lower efficiency of half body plan positive embryos in this experiment was due to the time taken to inject the large number of embryos (n=377, Table 1), which may have resulted in a large proportion being injected during the later stages of the first cell division, missing the optimal time window for early integration during or before the first cell division."

(2) Line 134 should not claim significance without statistical tests.

To avoid confusion with statistical significance we have changed the wording to describe the improvement from ~13% to ~92% as 'a dramatic improvement' rather than 'a significant improvement'.

Furthermore, the claim that gRNA incubation time improves activity (lines 130 to 136) cannot be compared across different CRISPR target sites, as every CRISPR paper since 2012 has shown that different spacers have different inherent activity.

We agree it is possible that we could not completely exclude that the dramatic improvement seen following the increased incubation time could be due to the 3 guides showing the higher rates being much more efficient than the 2 we tested at a shorter incubation. We therefore re-injected the same guides using the increased incubation times and saw improvements from 15% to 67% and 11% to 57%, demonstrating the increased frequency was due to the optimised protocol and not guide specific.

We have included this new data in Table S2 and added the following text to the manuscript "We additionally tested this improvement using the two *actc1b* targeting guideRNA's described above and saw dramatic improvements in integration (improving from 11% and 15% integration events to 57% and 67% respectively, and examples of fish with either half or the full body plan expression; Supplementary Table S2 & Fig. S1), alleviating any concerns that the improvement may be due to the guideRNA rather than the protocol."

Target gene (Intron)	Injected	Mosaic	Half or full body plan	Total positive
Protocol: Frozen guideRNA stock & 10 min incubation at 37°C				
actc1b-mTagBFP2-afpUTR (Intron 2)	148	20 (13.4%)	2 (1.3%)	22 (15%)
actc1b-mTagBFP2-sGFP1-10 (Intron 2)	93	12 (12.9%)	2 (2.2%)	14 (15.1%)
actc1b-mTagBFP2-sGFP11x7 (Intron 4)	377	42 (11%)	1 (0.2%)	43 (11.4%)

Protocol optimisation: Freshly prepared guideRNA & >30 min incubation at 37°C				
actc1b-mTagBFP2-afpUTR (Intron 2)	126	83 (66%)	1 (1%)	84 (67%)
actc1b-mTagBFP2-sGFP11x7 (Intron 4)	115	55 (48%)	11 (10%)	66 (57%)

Table S2. Injection efficiency of *actc1b* target sites before and after protocol optimisation. Targeting events below the dashed line are utilising the optimised protocol of freshly complexed guideRNA and an extended incubation time at 37 °C prior to injection.

The 30 minute incubation suggested here doesn't match the incubation time in the supplemental protocol (Figure S1 step 4).

We provided a window of time, which included 30 minutes, but as the reviewer has highlighted, we did not test all of the time periods in this window and so have amended this to be a >30 minute incubation.

If 30 minute incubation is apparently superior to shorter incubation, why was the suggested 1 hour incubation not tested?

There is potential that longer incubation periods would lead to further benefits but as we achieved mosaic embryos at a very high frequency, with every guide tested resulting in a higher frequency than the highest value we could find in previous literature and in half the guides tested ~95%, we decided the potential for further increases would be limited.

Furthermore, the authors should mention the cost-benefit of longer incubation with potential in vitro cleavage of the plasmid donor, as injection of linearized donor is generally not suitable for gene knockin.

Cas9 requires Mg²⁺ as a cofactor to cleave DNA. The injection mix does not contain Mg²⁺ and therefore the plasmid will not be linearised until injected. The initial 10 minute incubation is required for Cas9/guideRNA complex formation. The additional incubation period is to promote greater CRISPR/Cas9 binding to the targeting plasmid prior to injection.

We do not go into this level of detail in the text, but for readers who are interested in this level of detail we supplied a reference to the study that investigated the timing of the CRISPR/Cas9 mechanism.

(3) The authors emphasize that the contribution of this manuscript is the half-body fluorescence allowing for easier pre-screening of founder fish. In two of the seven lines tested, no half-body founders were observed. The authors should discuss what approaches are necessary to achieve these founders, and explain whether the method can generally be expected to work only 5/7 of the time.

We generated stable lines for all 7 targets in the manuscript. We did only see early integration events for 5 of the 7 lines and had included a possible explanation for this in the previous revision as the review requested. The reviewer stated was too speculative, and therefore, as discussed above we have modified this statement. In particular that section now ends with

“Therefore, we suggest injecting small batches of very recently laid embryos at a time to promote integration events before the cell completes its first cell division”.

Additional comments:

(1) One can easily overcome genetic compensation by inhibiting *upf3a* expression or disrupting *upf3a* gene when generating indel mutations of their target genes, and screening for indel mutations is much easier than screening for insertional mutations. The authors should also discuss the disadvantages of their method in preventing genetic compensation and why researchers should choose their method to avoid genetic compensation.

We disagree, we think screening for fluorescent animals as generated by the CRIMP approach is easier and cheaper than extracting DNA and then conducting PCR amplification followed by gel electrophoresis, melt curve, or PAGE analysis as would be required to identify indels generated using traditional CRISPR mutagenesis. It is also a lot more work to maintain and analyse mutant strains on a *upf3a* mutant background than individually.

(2) The authors note several embryos showing fluorescence in half of the body (line 157, 173). The only comparable micrograph shown (Fig 1E) is captioned as having fluorescence in "one half of the ventral body plan" (line 550). Half of the ventral body suggests one quarter of the whole body. Are the embryos not fluorescent if observed dorsally? The authors need to make clear what they observed.

The wording has been changed from “one half of the ventral body plan” to “one half of the body plan”, additionally we have provided images of fish from the new experiments added that demonstrate half body plan, and even complete body plan, expression in Supplementary Fig. S1.

Figure S1. Successful targeting events during the first cell division generate embryos with correct expression in one or both halves of the body plan. Optimised incubation times result in **(a)** correct integration of the targeting vector into intron 2 of *actc1b* during the first cell division as demonstrated by the embryo with mTagBFP2 expression in both sides of the body plan. In our experience, such embryos, when raised to adulthood and crossed, have always transmitted the successful integration event to their progeny. Integration at later stages generates embryos with mosaic pattern of expression. **(b)** Integration of the targeting vector into intron 4 of *actc1b* generates embryos with mTagBFP2 expression in both sides of the body plan, as well as embryos with expression in one side of the body plan. Both expression patterns demonstrate early integration events during the first cell division. Integration events later in development result in embryos displaying mTagBFP2 expression in a mosaic pattern. Ventral and dorsal views at 2 dpf. Scale bar indicates 500 μ m.

(3) Please provide the data of germline mosaicism of each positive founder fish in Table 1.

We have added germline transmission rates for the fish identified in Table 1 (where available) in Table S1.

Insertion line		Germline transmission
Ti(actc1b^{int2}-mTagBFP2)	Half body plan	21/54 (38%)
Ti(actc1b^{int4}-mTagBFP2-T2A-sGFP11x7)	Mosaic Half body plan	7/29 (19%) 49/158 (31%)
Ti(tdGF1^{int3}-Gal4vp16/4xnrUAS-mTagBFP2)	Half body plan	14/27 (52%)
Ti(vegfaa^{int1}-Gal4vp16_synCoTC/4xnrUAS-mTagBFP2)	Mosaic	12/51 (23.5%)

Table S1. Founder germline transmission rates. The germline transmission rate of successful targeting events for each target site.

(4) Any phenotype of the *actc1b* splitGFP mutant embryos?

No, the regulation of the actin paralogues is very complicated, and we see changes in the levels of other actins. We hope to explore this further in the future.

(5) Does *Ti(tdGF1^{int3}-Gal4vp16/4xnrUAS-mTagBFP2)* also show variable expression levels of mTagBFP2?

We did not observe variable expression with the *tdGF1* line.

(6) The Abstract is poorly written, did not emphasize the particular improvements in their methodology.

We apologise, we did not change the abstract since the initial submission but have now modified it to further emphasise the improvements the reviewer has identified.

“We developed a highly efficient targeted insertional mutagenesis system, CRIMP, and an associated plasmid toolkit, CRIMPkit, that disrupts native gene expression by inducing complete transcriptional termination, generating null mutant alleles without inducing genetic compensation. The protocol results in a high frequency of integration events and can generate very early targeted insertions, during the first cell division, producing embryos with expression in one or both halves of the body plan. Fluorescent readout of integration events facilitates selection of successfully mutagenized fish and, subsequently, visual identification of heterozygous and mutant animals. Together, these advances greatly improve the efficacy of generating and studying mutant lines. The CRIMPkit contains 24 ready-to-use plasmid vectors to allow easy and complete mutagenesis of any gene in any reading frame without requiring custom sequences, modification, or subcloning.”

(7) The manuscript is poorly organized, no Results section was found and the results were mixed with Introduction.

We have used the combined results and discussion format common in the journal, but the “Results and Discussion” section heading was absent, we have corrected this error.

Considering the above concerns, I think the authors should properly emphasize the

improvements on the mutagenesis methods by providing sufficient experimental evidence, besides introducing the design and generation of universal plasmids set. Otherwise the innovation of the manuscript is quite limited and it's more appropriate for this manuscript to publish as a resource paper (design and collections of the universal plasmids set) in a more specialized journal rather than NC.

We believe that where the reviewer has highlighted a lack of experimental evidence, specifically, using the same guides to confirm the improvement in the frequency of integration events, we have now supplied appropriate additional evidence to support our findings. The only other area where the reviewer suggested we did not have appropriate evidence was in the discussion around integration in the G2 phase of the cell cycle. In this case we have made it clearer that we are citing existing studies or have removed the hypothesis about the timing.

Reviewer #2 (Remarks to the Author):

The revised manuscript showed a large improvement, however, there are still many important issues need further clarification and/or improvements.

Major concerns:

1. What exactly does the CRIMP protocol mean? Since the authors have tested different conditions and make some improvements, they should summarize and clarify the final CRIMP protocol at the end, and indicate which are the crucial or necessary steps and conditions to achieve high integration efficiency and ensure successful germline transmission.

2. There are many conditions/steps described throughout the manuscript attributing to the CRIMP protocol, are all these conditions/steps necessary? The authors should clarify which are really essential to ensure high integration efficiency and make their method/protocol more practical and easy to follow, by providing proper control experiments. For example, are modified crRNA and tracrRNA really necessary for the improvement of the targeted integration and germline transmission efficiency? Purchasing modified crRNA and tracrRNA could be expensive and sometimes might not be easily and readily available (e.g., takes longer time). The authors should compare the effect of traditional home-made gRNA (by in vitro transcription) with the commercial modified crRNA and tracrRNA for their protocol.

3. The authors seem to believe Cas9 protein could facilitate earlier integration events than Cas9 mRNA, leading to higher integration efficiency. However, they did not provide experimental data supporting their point. In fact, there are paper which have discussed this issue (Albadri et al., *Methods*, 121–122 (2017): 77–85. <http://dx.doi.org/10.1016/j.ymeth.2017.03.005>) indicated that Cas9 mRNA behaved better than Cas9 protein in inducing NHEJ-based insertion, and this is also consistent with our experience, as well as other labs (through personal communications). Would be more convincing if the authors could design proper experiments to support their speculation.

4. The integration efficiency of the founder embryos injected by using the optimized protocol showed obvious variations. For example, the percentage of fluorescent positive founder embryos ranges from 57% to 96.1%, and the amount of founder embryos showing half or full body fluorescent pattern ranges from none to 11. Any explanation? Any trick/suggestion in the selection of target sites? Does the integration efficiency correlate with the mutagenic (indel) efficiency of the target sites? Would be more informative if the authors could provide a list of the indel mutagenic efficiency of each target site (i.e., the indel efficiency when only injecting the Cas9/gRNA complex without donor plasmid).

5. Lines 142-144: "Positive mTagBFP2 fluorescence was observed in 75% (90/120) of the injected embryos (Table 1) representing a dramatic improvement in the frequency of integration events and therefore this longer incubation was retained for all subsequent experiments". Here the claim of "improvement" is invalid since there is no proper control. How is the efficiency with injection of not freshly prepared gRNA complexes and less incubation time?

6. Lines 113-119: "Of the injected embryos 11.4% (43/377) displayed mTagBFP2 expression, and one embryo (0.2%) displayed the correct expression in half of the embryo body plan (Table 1). We suspect that the lower efficiency of half body plan positive embryos in this experiment was due to the time taken to inject the large number of embryos (n=377, Table 1), which may have resulted in a large proportion being injected during the later stages of the first cell division, missing the optimal time window for early integration during or before the first cell division." This speculation could be easily tested: separate the embryos into two groups, one from early injection and the other from late injection, and make a compare about their integration efficiency, fluorescent pattern, and germline transmission efficiency, etc.

7. The correlation between half-body fluorescent embryos and successful germline transmission is

quite impressive. The authors speculate early injection is important to generate such embryos; however, obtaining such embryos seems quite tricky and risky: (1) Not all the target sites/donor vectors could produce such embryos (2/9 failed, considering the data from Table1 and S2). (2) In most cases (6/7, considering the data from Table1 and S2) only 1 or 2 such embryos were observed from hundreds of injected embryos. The current data cannot fully support their speculation about the correlation since there is no proper comparison, and the authors could perform simple experiments to further evaluate their speculation, e.g., injecting embryos at different time points and compare the ratio of half-body fluorescent embryos and germline transmission.

8. Table 1: It's surprising that for *actc1b-mTagBFP2-afpUTR* insertion (before optimization of the experimental protocol), none of the founders derived from the 20 mosaic embryos showed successful germline transmission. Any explanation? Were these embryos derived from early injection? How "mosaic" were these embryos regarding the fluorescent pattern? In principle, if the integration indeed happened early (within or close to first cleavage), the mosaic pattern should be broad or close to half-body? Did the authors observe this tendency? Several publications have showed the impact of fluorescent patterns on germline transmission, where broad patterns are more likely to generate germline transmission events than sparse ones. The authors should pick up one or more previously reported target gene/target sites from the most relevant paper to compare the published method with their CRIMP protocol, to show the necessity or advantages of their protocol.

9. Table S1: These are not germline transmission rates but mosaicism of the germline of each founder. Mosaicism usually indicates the percentage of positive F1 embryos of a founder. Germline transmission rate usually indicates the percentage of founders that produce positive progeny. For example, according to Table 1, for *actc1b-mTagBFP2-afpUTR* insertion, the overall germline transmission rate is 0.7% (=1/148) since they authors identified one germline-positive founder from 148 founders, or 4.8% (=1/21) for the selected embryonic fluorescent founders since they authors identified one germline-positive founder from 21 founders derived from injected embryos showing fluorescent signals.

10. The optimized protocol seems to be advantageous in generating successful germline transmission. Which condition is crucial? Freshly prepared gRNA complexes, or >30 min incubation at 37°C?

11. Among the 20 founder fish derived from the mosaic embryos injected with *actc1b-mTagBFP2-afpUTR* using the un-optimized condition (i.e., frozen gRNA stock & 10 min incubation), none of them showed successful germline transmission, while only a founder derived from the half-body fluorescent embryo produced positive F1 embryos (Table 1). This suggests half-body plan pattern is crucial for successful germline transmission while mosaic pattern is much less likely to produce germline transmission, at least at this target site. However, injection of the same donor vector (*actc1b-mTagBFP2-afpUTR*) at the same target site with optimized protocol (i.e., freshly prepared gRNA & >30 min incubation time) produced more mosaic (66% vs 13.4%) embryos but less half-body fluorescent embryos (1 vs 2) (Table S2), which suggests this optimized protocol might not lead to better germline transmission rate since there is no increase in the percentage of half-body fluorescent embryos.

12. Did the 83 *actc1b-mTagBFP2-afpUTR* mosaic founders (Table S2) give rise to any germline transmission?

13. Injection of *actc1b-mTagBFP2-sGFP11x7* using un-optimized condition (i.e., frozen gRNA stock & 10 min incubation) only produced one half-body fluorescent embryo, the authors "suspect that the lower efficiency of half body plan positive embryos in this experiment was due to the time taken to inject the large number of embryos (n=377, Table 1), which may have resulted in a large proportion being injected during the later stages of the first cell division, missing the optimal time window for early integration during or before the first cell division" (lines 115-119). According to this speculation, the higher number (11) of half-body fluorescent *actc1b-mTagBFP2-sGFP11x7* embryos produced by using optimized condition (i.e., freshly prepared gRNA & >30 min incubation time) might be due to

less (=earlier) injected embryos (115) rather than optimized protocol (freshly prepared gRNA & >30 min incubation time)? Similar number of embryos and strictly controlled injection timing should be applied for proper comparison of these two protocols.

14. Are the 11 half or full body fluorescent actc1b-mTagBFP2-sGFP11x7 founders (Table S2) all give rise to germline transmission?

15. The authors suggest their optimized protocol (i.e., freshly prepared gRNA & >30 min incubation time) is better than the un-optimized one (i.e., frozen gRNA stock & 10 min incubation). However, they only compared these two protocols at two target sites (Table S2), where one site showed more (11, 10% vs 1, 0.2%) half or full body fluorescent embryos with optimized protocol, but the other one showed the opposite result, i.e., less (1, 1% vs 2, 1.3%) half or full body fluorescent embryos with the optimized protocol, though both showed higher mosaic integration (Table S2). Unfortunately, the authors did not provide germline transmission data of the two sites with the optimized protocol, according to their other data, half-body fluorescent embryos are more important to give rise to germline transmission, by this criteria the optimal protocol did not show consistent improvement since not both sites produced more half or full body fluorescent embryos. Apparently only these two examples are not sufficient to draw the conclusion that the optimized protocol is better. The improvement of the optimization of injection protocol is not sufficiently supported by the current data. More examples need to provide to demonstrate that freshly prepared gRNA and more incubation time is indeed beneficial.

16. One important advantage of the CRIMP protocol "is able to achieve a very high proportion of mosaic integrations (57-96%)" (line 286), and "is able to produce early integration events during the first cell division generating embryos with positive fluorescence throughout one or both halves of the embryonic body plan" (lines 288-290). However, since the aim of this protocol is to generate insertional mutation of the target gene (i.e., disruption of the target gene), high integration efficiency (means high mutation rate) might be harmful to the injected embryos if the target gene is essential for embryonic development. How is the survival rate of the injected embryos? Did the died embryos show broad pattern of fluorescent signal? It's possible that both alleles could be disrupted by the donor insertion during first zygotic cleavage (e.g., due to 100% integration), which might lead to lethality of the embryos when targeting an essential gene? Considering the half or full body plan embryos are rare (in most cases only 1 or 2 such embryos), is it possible that this phenomenon is due to this reason (i.e., the embryos with the highest insertion rate died)? If yes, this would be an intrinsic limitation of the CRIMP protocol.

17. Figure 2: (1) "f Fold change of mTagBFP2 mRNA levels compared to bag3 levels in bag3mTagBFP2 heterozygote and homozygote embryos." This sentence is confusing. Which "bag3 levels" were used for the comparison? Those in bag3mTagBFP2 heterozygote and homozygote embryos? But the bag3 expression is completely lost in the homozygote embryos and impossible to compare? How would the mTagBFP2 mRNA levels look like when normalized to the reference gene? (2) "rpl13 and ef1a were used as the reference genes." Why two reference genes were used for mRNA normalization? Did they give exactly the same results? Which reference gene was used in the two panels (e and f)?

18. The authors should also discuss the limitations of their method.

Minor concerns:

1. Naming of the transgenic lines are not consistent, for example, "Ti(actc1bint4-mTagBFP2-T2A-splitGFP11x7)" in main text, but "Ti(actc1bint4-mTagBFP2-T2A-sGFP11x7)" in the table.

2. According to Table 1 and S1, two independent Ti(actc1bint4-mTagBFP2-T2A-sGFP11x7) transgenic lines (from different founders) were identified (one from a half-body founder and the other from a mosaic founder), in principle they are different alleles since they were derived from two independent insertion events, and the (5' and 3') junction sequences between the endogenous genomic site and

the inserted plasmid are most likely different due to different indels produced during insertion process. However, the authors used the same name for these two different lines, which is confusing. Separate names should be assigned to these two lines.

3. Lines 121-125: "Embryos displaying expression in half the body plan were raised to adulthood and outcrossed to wildtype fish to identify founders. All individuals that displayed such expression demonstrated germline transmission of the targeted insertion with the correct fluorophore expression pattern corresponding to actc1b." Are these sentences refer to the generation of actc1bsGFP1 and actc1bsGFP2 lines? There were two half-body founder embryos identified from pSA0-mTagBFP2-T2A-splitGFP1-10 injected embryos, but only one was raised to adult according to Table 1, which is inconsistent with the above claim that "Embryos displaying expression in half the body plan were raised to adulthood."

4. Line 126: "Crossing the two actc1bsGFP lines together at the F2 generation": There is no such lines called actc1bsGFP (should be either actc1bsGFP1 or actc1bsGFP2), the authors should be precise when talking about transgenic lines.

5. Table 1, Table S2: "Target gene" label in the first column is incorrect, the names (e.g, actc1b-mTagBFP2-afpUTR) under this column rather look like the donor vector but not the name of the target gene itself.

6. The information in Table 1 and Table S2 are largely overlapping, and Table S2 contains important information showing the improvement of the CRIMP protocol, would be more informative to combine these two tables into one.

7. Table 1: How many mosaic embryos were identified from actc1b-mTagBFP2-afpUTR injection? 20 (left side) or 22 (right side)?

8. Recently, there is a publication describing optimization of gene tagging strategies for illuminating expression profiles of genes with different abundance in zebrafish (Liu et al., Communications Biology, (2023) 6: 1300. <https://doi.org/10.1038/s42003-023-05686-1>), the authors should cite this paper and give a discussion.

We have responded to each of the comments from Reviewer 2 below. We appreciate the interest of reviewer 2 in the finer details of the methodology, however, believe that exploring the efficacy of different reagents and subtle changes in timing are outside the scope of the research and that the comments do not raise issues that affect the major outcomes of the manuscript.

REVIEWER COMMENTS

Reviewer #2 (Remarks to the Author):

The revised manuscript showed a large improvement, however, there are still many important issues need further clarification and/or improvements.

Major concerns:

1. What exactly does the CRIMP protocol mean? Since the authors have tested different conditions and make some improvements, they should summarize and clarify the final CRIMP protocol at the end, and indicate which are the crucial or necessary steps and conditions to achieve high integration efficiency and ensure successful germline transmission.

The description of CRIMP is in the introduction.

“CRISPR/Cas9 Insertional Mutagenesis Protocol (CRIMP) and toolkit (CRIMPkit)”

We have described an approach to create mutant lines using insertional mutagenesis. Whilst this has been described before, our approach does not require homology domains in the inserted sequences, removing the need to customise this for each target gene and allowing us to provide a suite of vectors (the toolkit) allowing any gene to be targeted, with multiple options for different fluorophores and amplification.

We additionally demonstrate the value of including a transcriptional termination sequence in the vectors, ensuring the endogenous transcript cannot be formed and removing variability in the expression of the inserted transgene.

Finally, we include details of our protocol, including freshly prepared guides, pre-hybridisation of the Cas9 protein, and injection into recently fertilised embryos, allowing integration at the first cell division. This results in embryos for which half, or the entire body plan contains the inserted sequence, these embryos always transmit to the next generation removing the need for time consuming screening, and along with the

universal nature of the toolkit greatly simplifies the process of generating mutant lines. These early integration events have not been achieved by other protocols and are also a significant advance.

The approach and protocol have not changed since the first submission of the manuscript, and we don't believe it requires further clarification now in the 4th version.

The protocol is described in the manuscript text, a step-by-step protocol suitable for use in the lab is provided in the supplementary data. We believe all the steps included are necessary.

2. There are many conditions/steps described throughout the manuscript attributing to the CRIMP protocol, are all these conditions/steps necessary? The authors should clarify which are really essential to ensure high integration efficiency and make their method/protocol more practical and easy to follow, by providing proper control experiments. For example, are modified crRNA and tracrRNA really necessary for the improvement of the targeted integration and germline transmission efficiency? Purchasing modified crRNA and tracrRNA could be expensive and sometimes might not be easily and readily available (e.g., takes longer time). The authors should compare the effect of traditional home-made gRNA (by in vitro transcription) with the commercial modified crRNA and tracrRNA for their protocol.

We believe all the steps outlined in the protocol are necessary.

We have developed an approach, toolkit, and a protocol as described in the manuscript. We have not evaluated all the possible options at every step and do not consider this necessary. It is of course something that others could investigate in the future if they felt this would be beneficial.

We use guideRNA's designed by Integrated DNA technologies (IDT) in all our CRISPR/Cas9 applications, in our experience they have been highly efficient, storage stable, with none of the toxicity that can be associated with home-made gRNA (by in vitro transcription) as outlined in Amorim et al., (2020). These were the reagents we have used to generate the results presented; we do not believe it is necessary to test all alternatives. Others interested in applying the approach can of course modify the protocol as they wish.

Amorim, J. P., Bordeira-Carriço, R., Gali-Macedo, A., Perrod, C., & Bessa, J. (2020). CRISPR-Cas9-Mediated Genomic Deletions Protocol in Zebrafish.

STAR Protocols, 1(3), 100208.

<https://doi.org/https://doi.org/10.1016/j.xpro.2020.100208>

3. The authors seem to believe Cas9 protein could facilitate earlier integration events than Cas9 mRNA, leading to higher integration efficiency. However, they did not provide experimental data supporting their point. In fact, there are paper which have discussed this issue (Albadri et al., *Methods*, 121–122 (2017): 77–85.

<http://dx.doi.org/10.1016/j.ymeth.2017.03.005>) indicated that Cas9 mRNA behaved better than Cas9 protein in inducing NHEJ-based insertion, and this is also consistent with our experience, as well as other labs (through personal communications). Would be more convincing if the authors could design proper experiments to support their speculation.

There are multiple papers that have investigated this issue and whilst the manuscript the reviewer refers to does identify slightly improved integration efficiency at later stages with mRNA in their hands, there is a consistent agreement that Cas9 Protein has higher cutting efficiencies, and cuts earlier than injected Cas9 mRNA including the results of Albadhri et al., referenced by the reviewer, and the other studies referenced in the background of that manuscript.

Critically, however, none of these studies identify integration at the first cell division, as we do. Our approach uses non-homologous end joining which is active at the first cell division and then becomes much less active for the next 12 cell divisions. This is well described in the literature, again, including the Albadhri manuscript. We include a figure from that manuscript below.

[Redacted]

Speed is critical for this early integration to occur, we inject a pre-formed complex of Cas9 protein, guide RNA and DNA into freshly fertilised embryos to achieve integration in the narrow time window (and achieve high efficiency at later, slower cell divisions as well).

If we were to inject Cas9 mRNA and wait for transcript to occur and an active complex to form the critical time window would be missed and we don't believe it is necessary to demonstrate that Cas9 protein and a pre-formed Cas9 complex would function earlier than one generated by injected mRNA. This is clearly established in the literature and retesting the benefits of protein vs mRNA is not justified.

4. The integration efficiency of the founder embryos injected by using the optimized protocol showed obvious variations. For example, the percentage of fluorescent positive founder embryos ranges from 57% to 96.1%, and the amount of founder embryos showing half or full body fluorescent pattern ranges from none to 11. Any explanation? Any trick/suggestion in the selection of target sites? Does the integration efficiency correlate with the mutagenic (indel) efficiency of the target sites? Would be more informative if the authors could provide a list of the indel mutagenic efficiency of each target site (i.e., the indel efficiency when only injecting the Cas9/gRNA complex without donor plasmid).

Reviewer #2 is incorrect in attributing the positive fluorescent ranges of 57% to 96.1% to positive founder embryos. These values on the table correspond to the proportion of injected embryos that have positive integrations that generate fluorescence, not screened founders.

In the current manuscript we already discussed this, as the reviewer previously asked us to speculate as to the cause, and provide an explanation for differences in the timing of injection and the requirement to inject embryos soon after being laid to achieve early integration events required to generate half-body positive embryos: *"We suspect that the lower efficiency of half body plan positive embryos in this experiment was due to the time taken to inject the large number of embryos (n=377, Table 1), which may have resulted in a large proportion being injected during the later stages of the first cell division, missing the optimal time window for early integration during or before the first cell division. Therefore, we suggest injecting small batches of very recently laid embryos at a time to promote integration events before the cell completes its first cell division."*

It is also worth noting that this variation ranges from the highest value previously reported in the literature (57%) up to 96%, substantially improving on the previous studies.

We have already provided the guidelines on target site selection in the methods section: *“Intronic guideRNA sites in target genes were identified using the IDTTM online tool.....target sites with the highest on-target and off-target scores were selected”*

Reviewer #2's suggestion to conduct further experiments to determine the mutagenic efficiency of each target site in the absence of the target vector is unnecessary for our study. The robust fluorescent readout of positive integration, half-positive embryos, and the successful generation of stable transgenic lines already demonstrate high cutting activity. It is not clear how these additional experiments would benefit the manuscript or the community that would apply the approach we describe.

5. Lines 142-144: “Positive mTagBFP2 fluorescence was observed in 75% (90/120) of the injected embryos (Table 1) representing a dramatic improvement in the frequency of integration events and therefore this longer incubation was retained for all subsequent experiments”. Here the claim of “improvement” is invalid since there is no proper control. How is the efficiency with injection of not freshly prepared gRNA complexes and less incubation time?

75% is clearly a substantial improvement on the 15% rate we initially achieved. However, as the reviewer highlighted in their previous review, we did not initially test the same guide in both conditions. We did therefore, in the previous revision, repeat injections with the two *actc1b* targeting guides using the improved conditions. These experiments are described in lines 145-149 and clearly show an improvement.

6. Lines 113-119: “Of the injected embryos 11.4% (43/377) displayed mTagBFP2 expression, and one embryo (0.2%) displayed the correct expression in half of the embryo body plan (Table 1). We suspect that the lower efficiency of half body plan positive embryos in this experiment was due to the time taken to inject the large number of embryos (n=377, Table 1), which may have resulted in a large proportion being injected during the later stages of the first cell division, missing the optimal time window for early integration during or before the first cell division.” This speculation could be easily tested: separate the embryos into two groups, one from early injection and the other from late injection, and make a compare about their integration efficiency, fluorescent pattern, and germline transmission efficiency, etc.

In our experience missing the early injection time window never results in early integration events, we have clearly discussed this in the manuscript. Furthermore, this is evidenced by examples where we did not observe early integration events but were able to generate lines from mosaic embryos. We believe that the proposed experiments of injecting after a delay to not generate half positive embryos, does not provide any additional benefit to the manuscript or the protocol we present. As discussed above the theory behind the early integrations is well established.

7. The correlation between half-body fluorescent embryos and successful germline transmission is quite impressive. The authors speculate early injection is important to generate such embryos; however, obtaining such embryos seems quite tricky and risky: (1) Not all the target sites/donor vectors could produce such embryos (2/9 failed, considering the data from Table1 and S2). (2) In most cases (6/7, considering the data from Table1 and S2) only 1 or 2 such embryos were observed from hundreds of injected embryos. The current data cannot fully support their speculation about the correlation since there is no proper comparison, and the authors could perform simple experiments to further evaluate their speculation, e.g., injecting embryos at different time points and compare the ratio of half-body fluorescent embryos and germline transmission.

We disagree that there is a risk or challenge associated with generating these early integrations. They do occur at low frequencies, but it is highly feasible to inject several hundred embryos in a single morning. In the cases where early integration does not occur, germline transmission was still achieved, but we believe it is more time effective to repeat injections if necessary and avoid raising and time-consuming screening of many potential founder fish, and instead raise a few fish with a guarantee of germline transmission.

The only way to get the entire body plan or 50% of the body plan to express the inserted transgene, as we do, is through an early integration event at the first cell division. This is not only logical but supported by the literature as discussed above. However, as we do not have data to directly examine this, we have therefore appropriately phrased this as speculation. We do not believe it is necessary to carry out injections at a later time point to demonstrate this.

In addressing our 'speculation' that the half-body fluorescent embryos successfully transmit the insertions to the next generations, this is not speculation, we have seen transmission in every case examined. Furthermore, this is exactly what would be expected when you see transgene expression throughout one half of an embryo.

8. Table 1: It's surprising that for actc1b-mTagBFP2-afpUTR insertion (before optimization of the experimental protocol), none of the founders derived from the 20 mosaic embryos showed successful germline transmission. Any explanation? Were these embryos derived from early injection? How "mosaic" were these embryos regarding the fluorescent pattern? In principle, if the integration indeed happened early (within or close to first cleavage), the mosaic pattern should be broad or close to half-body? Did the authors observe this tendency? Several publications have showed the impact of fluorescent patterns on germline transmission, where broad patterns are more likely to generate germline transmission events than sparse ones. The authors should pick up one or more previously reported target gene/target sites from the most relevant paper to compare the published method with their CRIMP protocol, to show the necessity or advantages of their protocol.

All of the founders identified showed germline transmission. The fish the reviewer refers to are not founders but F₀ injected fish raised to adulthood.

These fish showed mosaic expression as embryos and had a relatively small number of positive fibres. The failure to identify a founder from amongst these fish is not unusual and helps to illustrate the benefit of early integration events.

It would be helpful if the reviewer supplied references to support their statement about expression pattern influencing germline transmission. However, we believe that broader expression patterns make the identification of transgenic fish easier but does not affect the chance of an early integration event.

Reviewer #2 has suggested that if early integration events occur that broader patterns close to half-body should be observed in mosaic embryos. This is not correct and highlights a misunderstanding of the timing of integration events. Integration is clearly possible at the first cell division as evidenced by half and full body positive embryos we have identified. Integration would be at later development stages, and result in the typical mosaic pattern we observe and as reported in the literature, that gives 'mosaic' fish their name.

As discussed in the manuscript, if integration does not occur during the first cell division of an embryo, it is only possible to integrate after the 1000 cell stage when the cell division lengthens and acquires a G phase, generating mosaic embryos, which are less likely to transmit to the next generation. This has been previously discussed in the

literature and is supported by the fact we have seen many half or full body plan positive fish but never a 25% or 12.5% etc.

We do not understand the purpose of the proposed experiments.

9. Table S1: These are not germline transmission rates but mosaicism of the germline of each founder. Mosaicism usually indicates the percentage of positive F1 embryos of a founder. Germline transmission rate usually indicates the percentage of founders that produce positive progeny. For example, according to Table 1, for *actc1b-mTagBFP2-afpUTR* insertion, the overall germline transmission rate is 0.7% (=1/148) since they authors identified one germline-positive founder from 148 founders, or 4.8% (=1/21) for the selected embryonic fluorescent founders since they authors identified one germline-positive founder from 21 founders derived from injected embryos showing fluorescent signals.

We disagree with the reviewers' definitions and find them confusing, as mosaicism is not present in the F1 animals, but we would be happy to retitle the column in Table S1 and alter the legend to make clear we are looking at the proportion of F1 embryos that have inherited the integration event from the founder.

The column title now reads:

"Proportion of positive F1 embryos"

The legend now reads:

"The proportion of F1 embryos that inherited successful integration events from a founder for each target site."

In suggesting that the germline transmission rate is 0.7% Reviewer #2 has misinterpreted the data represented in Table1. 148 embryos were injected, only those showing positive expression were raised to adulthood and screened. Of the half-body positive fish the germline transmission rate was 100%, for the mosaic fish it was indeed much lower, the full numbers are presented in the table.

10. The optimized protocol seems to be advantageous in generating successful germline transmission. Which condition is crucial? Freshly prepared gRNA complexes, or >30 min incubation at 37°C?

We have included both as they both contribute to the success. The protocol does not include steps or conditions that we believe are unnecessary.

11. Among the 20 founder fish derived from the mosaic embryos injected with *actc1b-mTagBFP2-afpUTR* using the un-optimized condition (i.e., frozen gRNA stock & 10 min incubation), none of them showed successful germline transmission, while only a founder derived from the half-body fluorescent embryo produced positive F1 embryos (Table 1). This suggests half-body plan pattern is crucial for successful germline transmission while mosaic pattern is much less likely to produce germline transmission, at least at this target site. However, injection of the same donor vector (*actc1b-mTagBFP2-afpUTR*) at the same target site with optimized protocol (i.e., freshly prepared gRNA & >30 min incubation time) produced more mosaic (66% vs 13.4%) embryos but less half-body fluorescent embryos (1 vs 2) (Table S2), which suggests this optimized protocol might not lead to better germline transmission rate since there is no increase in the percentage of half-body fluorescent embryos.

As above the reviewer is incorrectly referring to adult fish as founders, this should be reserved for fish which have integrated the transgenic construct.

While the rate of half-body positive embryos at this target site (*actc1b* intron-2) using the optimised conditions was slightly reduced (1% vs 1.6%), we do not agree that such a small difference indicates a significant reduction. Furthermore, at the second target site tested (*actc1b* intron-4) the rate of half-positive embryos increased by 9.8% (0.2% vs 10%). No other protocol has detected any such early events - both 1% and 10% are excellent achievements. When examining the change in mosaic expressing fish - these improve from 15% and 11% to 57% and 67%, further supporting the improvement.

12. Did the 83 *actc1b-mTagBFP2-afpUTR* mosaic founders (Table S2) give rise to any germline transmission?

Reviewer #2's referral to the 83 embryos *actc1b-mTagBFP2-afpUTR* embryos as founders in Table S2 is incorrect. Founder's inherently having to show germline transmission to be considered founders. The data and text in Table S2 clearly describes the percentage of injected embryos that displayed positive fluorescence. This data was generated and included at Reviewer #2's request during the last submission round. These fish were not raised, as this line had already been successfully established and the purpose of the experiment was to test the improvement to the protocol in generating fluorophore expressing F₀ fish.

13. Injection of actc1b-mTagBFP2-sGFP11x7 using un-optimized condition (i.e., frozen gRNA stock & 10 min incubation) only produced one half-body fluorescent embryo, the authors “suspect that the lower efficiency of half body plan positive embryos in this experiment was due to the time taken to inject the large number of embryos (n=377, Table 1), which may have resulted in a large proportion being injected during the later stages of the first cell division, missing the optimal time window for early integration during or before the first cell division” (lines 115-119). According to this speculation, the higher number (11) of half-body fluorescent actc1b-mTagBFP2-sGFP11x7 embryos produced by using optimized condition (i.e., freshly prepared gRNA & >30 min incubation time) might be due to less (=earlier) injected embryos (115) rather than optimized protocol (freshly prepared gRNA & >30 min incubation time)? Similar number of embryos and strictly controlled injection timing should be applied for proper comparison of these two protocols.

The key outcome of our study is a highly efficient, universally applicable, mutagenesis approach that can generate very early integration events. The reviewer previously asked us to speculate as to why we did not see more half body plan embryos and we have provided a reasonable hypothesis. The data from the four other guides used clearly demonstrates the benefit of the optimised conditions. We believe early injection is also important for the reasons outlined above. It is not of benefit to dissect what level of improvement is attributable to the different aspects of the protocol.

We have already provided a protocol that can generate targeting events at stages earlier than has been previously reported, and provided what we believe to be the best methods for achieving this. Reviewer #2’s request to conduct further experiments that provide no practical benefit to the target audience.

14. Are the 11 half or full body fluorescent actc1b-mTagBFP2-sGFP11x7 founders (Table S2) all give rise to germline transmission?

The data in Table S2 was generated during the last revision round at Reviewer #2’s request. These embryos were not raised to adulthood and screened, as this was not required to address the issue the reviewer previously raised. All previous half-body positive embryos screened to date were identified as founders, and we if there had been time to raise these, and we had done so they would have all passed on the integration to the next generation. However, as this was not tested, we have not referred to them as founders.

15. The authors suggest their optimized protocol (i.e., freshly prepared gRNA & >30 min incubation time) is better than the un-optimized one (i.e., frozen gRNA stock & 10 min incubation). However, they only compared these two protocols at two target sites (Table S2), where one site showed more (11, 10% vs 1, 0.2%) half or full body fluorescent embryos with optimized protocol, but the other one showed the opposite result, i.e., less (1, 1% vs 2, 1.3%) half or full body fluorescent embryos with the optimized protocol, though both showed higher mosaic integration (Table S2). Unfortunately, the authors did not provide germline transmission data of the two sites with the optimized protocol, according to their other data, half-body fluorescent embryos are more important to give rise to germline transmission, by this criteria the optimal protocol did not show consistent improvement since not both sites produced more half or full body fluorescent embryos. Apparently only these two examples are not sufficient to draw the conclusion that the optimized protocol is better. The improvement of the optimization of injection protocol is not sufficiently supported by the current data. More examples need to provide to demonstrate that freshly prepared gRNA and more incubation time is indeed beneficial.

This is revisiting an earlier point again. We have tested 5 guides with the optimised protocol. We directly compared two to the original protocol and show clear improvement, and for all 5 guides achieved a frequency of positive embryos that matches or improves on the highest frequency previously described in the literature. Together this data clearly demonstrates that the protocol is effective.

We believe the improvements are clear, and even without the improvement in the preparation of reagents the achievements of the study are a sufficient advance on existing mutagenesis approaches.

16. One important advantage of the CRIMP protocol “is able to achieve a very high proportion of mosaic integrations (57-96%)” (line 286), and “is able to produce early integration events during the first cell division generating embryos with positive fluorescence throughout one or both halves of the embryonic body plan” (lines 288-290). However, since the aim of this protocol is to generate insertional mutation of the target gene (i.e., disruption of the target gene), high integration efficiency (means high mutation rate) might be harmful to the injected embryos if the target gene is essential for embryonic development. How is the survival rate of the injected embryos? Did the died embryos show broad pattern of fluorescent signal? It’s possible that both alleles could be disrupted by the donor insertion during first zygotic cleavage (e.g., due to 100% integration), which might lead to lethality of the embryos when targeting an essential

gene? Considering the half or full body plan embryos are rare (in most cases only 1 or 2 such embryos), is it possible that this phenomenon is due to this reason (i.e., the embryos with the highest insertion rate died)? If yes, this would be an intrinsic limitation of the CRIMP protocol.

We appreciate that the reviewer is suggesting the approach is now so efficient that it may be a limitation.

We have high survival rates for our embryos, especially our half-positive embryos as indicated in Table 1. Due to the high rate of survival and lack of phenotypes in these animals as adults, we believe the half-body or full-body positive embryos described in this paper represent heterozygous insertion in half of the body, and therefore do not generate half-positive embryos that are mutant on the positive side.

If the high efficiency was a problem in some instances, the improvements to the protocol could be removed to reduce efficiency.

17. Figure 2: (1) “f Fold change of mTagBFP2 mRNA levels compared to bag3 levels in bag3mTagBFP2 heterozygote and homozygote embryos.” This sentence is confusing. Which “bag3 levels” were used for the comparison? Those in bag3mTagBFP2 heterozygote and homozygote embryos? But the bag3 expression is completely lost in the homozygote embryos and impossible to compare? How would the mTagBFP2 mRNA levels look like when normalized to the reference gene? (2) “rpl13 and ef1α were used as the reference genes.” Why two reference genes were used for mRNA normalization? Did they give exactly the same results? Which reference gene was used in the two panels (e and f)?

In Figure 2f the comparison is between mTagBFP2 expression levels and the levels of *bag3* in the wildtype siblings.

We have rephrased the legend so it now reads:

“Fold change of mTagBFP2 mRNA levels in bag3^{mTagBFP2} heterozygote and homozygote embryos compared to bag3 mRNA levels in wildtype siblings.”

The use of two (or more) reference genes (geomean) is common practice in qRT-PCR and is considered best practice and in keeping with best practice we used the geomean of the two reference genes.

18. The authors should also discuss the limitations of their method.

For genes only expressed at adult stages visual selection would not be possible but we do not believe this warrant discussion. Other than suggesting the high efficiency of our approach may be a limitation above, the reviewer has not suggested any limitation in the latest comment, and we have address previous comments about suitability for low expressing genes in previous reviews by highlighting that we have successfully applied the approach to genes with even lower levels of expression than the examples the reviewer suggested.

Minor concerns:

1. Naming of the transgenic lines are not consistent, for example, “Ti(actc1b^{int4}-mTagBFP2-T2A-splitGFP11x7)” in main text, but “Ti(actc1b^{int4}-mTagBFP2-T2A-sGFP11x7)” in the table.

We have updated Figure 1 to read:

“Ti(actc1b^{int2}-mTagBFP2-T2A-splitGFP1-10)” and “Ti(actc1b^{int4}-mTagBFP2-T2A-splitGFP11x7)”

And have updated Table S1 to read:

“Ti(actc1b^{int4}-mTagBFP2-T2A-splitGFP11x7)”

2. According to Table 1 and S1, two independent Ti(actc1b^{int4}-mTagBFP2-T2A-sGFP11x7) transgenic lines (from different founders) were identified (one from a half-body founder and the other from a mosaic founder), in principle they are different alleles since they were derived from two independent insertion events, and the (5' and 3') junction sequences between the endogenous genomic site and the inserted plasmid are most likely different due to different indels produced during insertion process. However, the authors used the same name for these two different lines, which is confusing. Separate names should be assigned to these two lines.

We have added ‘founder a’ and ‘founder b’ to Table S1.

3. Lines 121-125: “Embryos displaying expression in half the body plan were raised to adulthood and outcrossed to wildtype fish to identify founders. All individuals that displayed such expression demonstrated germline transmission of the targeted insertion with the correct fluorophore expression pattern corresponding to actc1b.” Are these sentences refer to the generation of actc1bsGFP1 and actc1bsGFP2 lines? There were two half-body founder embryos identified from pSA0-mTagBFP2-T2A-splitGFP1-10 injected embryos, but only one was raised to adult according to Table 1, which is

inconsistent with the above claim that “Embryos displaying expression in half the body plan were raised to adulthood.”

This is incorrect, table 1 shows a single half body plan positive fish with this construct.

4. Line 126: “Crossing the two *actc1b*sGFP lines together at the F2 generation”: There is no such lines called *actc1b*sGFP (should be either *actc1b*sGFP1 or *actc1b*sGFP2), the authors should be precise when talking about transgenic lines.

The text has been updated to read:

*“Crossing *actc1b*^{sGFP1} and *actc1b*^{sGFP2} lines together at the F2 generation”*

5. Table 1, Table S2: “Target gene” label in the first column is incorrect, the names (e.g, *actc1b*-mTagBFP2-afpUTR) under this column rather look like the donor vector but not the name of the target gene itself.

The column heading of Table 1 and Table S2 have been changed to:

“Target gene-construct”

6. The information in Table 1 and Table S2 are largely overlapping, and Table S2 contains important information showing the improvement of the CRIMP protocol, would be more informative to combine these two tables into one.

Table S2 was to address the reviewer’s previous comment on the need to address the proportion of positive fish with the original and improved protocol for the reagent preparation and we do not have the corresponding adult data as in table 1. Therefore, it is not appropriate to merge them.

7. Table 1: How many mosaic embryos were identified from *actc1b*-mTagBFP2-afpUTR injection? 20 (left side) or 22 (right side)?

We have confirmed that 20 is correct and have corrected the typo in Table 1. Numbers in the rest of the manuscript are unaffected.

8. Recently, there is a publication describing optimization of gene tagging strategies for illuminating expression profiles of genes with different abundance in zebrafish (Liu et al.,

Communications Biology, (2023) 6: 1300. <https://doi.org/10.1038/s42003-023-05686-1>), the authors should cite this paper and give a discussion.

This an additional paper from one of the teams whose work the reviewer has championed and suggested for inclusion in previous comments. Our manuscript describes an intronic insertional mutagenesis approach that is highly efficient, does not require customisation for each target gene, and a toolkit to facilitate the adoption of the approach. It also includes additional refinements such as the inclusion of transcription termination sequences, a split-fluorophore system for genotyping, and we describe, for the first time, early integration events that dramatically improve the efficiency of mutant line generation.

The paper the reviewer asks us to cite describes the use of a micro-homology-based approach that targets exons, requiring precise repair, customisation for each new target gene, and is focussed on tagging rather than mutagenesis. We don't believe that it aligns sufficiently with the work conducted to warrant discussion.